# Biogenesis of phased siRNAs on membrane-bound polysomes in Arabidopsis

Shengben Li[1,2†], Brandon Le[2†], Xuan Ma[3,4†], Shaofang Li[2†], Chenjiang You[2,3], Yu Yu[2], Bailong Zhang[2], Lin Liu[2,3], Lei Gao[2,3], Ting Shi[2,5], Yonghui Zhao[2], Beixin Mo[3], Xiaofeng Cao[4], Xuemei Chen[2,3,6*]

[1]Agricultural Genomics Institute, Chinese Academy of Agricultural Sciences, Shenzhen, China; [2]Department of Botany and Plant Sciences, Institute of Integrative Genome Biology, University of California, Riverside, Riverside, United States; [3]Guangdong Provincial Key Laboratory for Plant Epigenetics, College of Life Sciences and Oceanography, Shenzhen University, Shenzhen, China; [4]State Key Laboratory of Plant Genomics and National Center for Plant Gene Research, Institute of Genetics and Developmental Biology, Beijing, China; [5]College of Horticulture, Nanjing Agricultural University, Nanjing, China; [6]Howard Hughes Medical Institute, University of California, Riverside, Riverside, United States

*For correspondence: xuemei.chen@ucr.edu

[†]These authors contributed equally to this work

**Abstract** Small RNAs are central players in RNA silencing, yet their cytoplasmic compartmentalization and the effects it may have on their activities have not been studied at the genomic scale. Here we report that Arabidopsis microRNAs (miRNAs) and small interfering RNAs (siRNAs) are distinctly partitioned between the endoplasmic reticulum (ER) and cytosol. All miRNAs are associated with membrane-bound polysomes (MBPs) as opposed to polysomes in general. The MBP association is functionally linked to a deeply conserved and tightly regulated activity of miRNAs – production of phased siRNAs (phasiRNAs) from select target RNAs. The phasiRNA precursor RNAs, thought to be noncoding, are on MBPs and are occupied by ribosomes in a manner that supports miRNA-triggered phasiRNA production, suggesting that ribosomes on the rough ER impact siRNA biogenesis. This study reveals global patterns of cytoplasmic partitioning of small RNAs and expands the known functions of ribosomes and ER.

## Introduction

RNA silencing is a conserved posttranscriptional gene regulatory mechanism in eukaryotes. At the core of RNA silencing are small RNAs, microRNAs (miRNAs) or small interfering RNAs (siRNAs), and their effector ARGONAUTE (AGO) proteins. Many observations document the membrane association of plant and animal AGO proteins (*Brodersen et al., 2012*; *Cikaluk et al., 1999*; *Gibbings et al., 2009*; *Jouannet et al., 2012*; *Lee et al., 2009*; *Li et al., 2013*; *Stalder et al., 2013*; *Wu et al., 2013*), but the cytoplasmic location where miRNAs or siRNAs repress target RNAs is largely unexplored and is perhaps presumed to be the cytosol. The membrane-cytosol partitioning of small RNAs has not been studied at the genomic scale. Little is known about how the membrane-cytosol partitioning of small RNAs affects their activities.

A well-documented feature of RNA silencing in plants and *C. elegans* is signal amplification, in which primary siRNAs from transgenes or viruses guide the production of secondary siRNAs from target RNAs through the activities of RNA-dependent RNA polymerases (RdRP) (*Sijen et al., 2001*;

*Vaistij et al., 2002*). This results in enhanced silencing and the spreading of the silencing signal into flanking sequences. Contrary to siRNAs, most miRNAs do not cause 'signal amplification' from their target RNAs. In fact, should miRNAs do so, the secondary siRNAs may cause unintended repression of genes homologous to miRNA targets.

However, for a small number of miRNAs and their targets, such 'signal amplification' occurs in plants. In fact, a widespread and deeply conserved phenomenon in diverse land plants is miRNA-triggered production of phased secondary siRNAs (phasiRNAs) from target transcripts (reviewed in [*Fei et al., 2013*]). Upon miRNA-guided cleavage of a target RNA, either the 5' or 3' cleavage fragment is converted by RdRP to double-stranded RNA, which is then successively diced into 21-nt phasiRNAs, with the phase determined by miRNA-guided cleavage (*Allen et al., 2005*; *Axtell et al., 2006*). In *Arabidopsis*, eight noncoding *TAS* loci generate phasiRNAs in this manner and, as the phasiRNAs regulate target genes in trans, they are termed trans-acting siRNAs (ta-siRNAs) (*Allen et al., 2005*; *Axtell et al., 2006*; *Peragine et al., 2004*; *Rajagopalan et al., 2006*; *Vazquez et al., 2004b*). Most miRNAs are 21 nt long and do not trigger phasiRNA production. A predominant mechanism (termed the 'one hit model') to trigger phasiRNA production is by a 22-nt miRNA (*Chen et al., 2010*; *Cuperus et al., 2010*); the *TAS1*, *2*, and *4* loci are examples of the 'one-hit' model with 22-nt miR173 or miR828 as the trigger (*Allen et al., 2005*; *Rajagopalan et al., 2006*). In a 'two-hit' model, a pair of 21-nt miRNAs target the same transcript to cause phasiRNA production (*Axtell et al., 2006*). The three *TAS3* loci are such examples, each containing two sites for miR390 (*Axtell et al., 2006*), a miRNA that associates with AGO7 instead of AGO1 to trigger phasiRNA production (*Montgomery et al., 2008*). In addition to the noncoding *TAS* loci, a small number of protein-coding genes in *Arabidopsis* such as immune receptor (*NBS-LRR*) and pentatricopeptide repeat (*PPR*) genes are targeted by 22-nt miRNAs and produce phasiRNAs (*Chen et al., 2007*; *Howell et al., 2007*).

Where phasiRNA biogenesis occurs in the cytoplasm is unknown. Intriguingly, SGS3 and RDR6, two proteins required for phasiRNA biogenesis (*Peragine et al., 2004*; *Vazquez et al., 2004b*), form cytoplasmic foci called siRNA bodies, which are often adjacent to vesicles marked by a cis-Golgi marker (*Jouannet et al., 2012*). In addition, both SGS3 and AGO7 are present in the microsomal fraction (*Jouannet et al., 2012*), implicating that ta-siRNA biogenesis occurs on a cytoplasmic membrane structure. A recent study found that *TAS3* RNA is bound by ribosomes, and ribosomes on *TAS3* RNA are stalled by AGO7 at the miR390-binding site, implicating that ta-siRNA biogenesis from *TAS3* occurs on polysomes (*Hou et al., 2016*).

We combined genomic approaches with cellular fractionation to study the subcellular distribution of small RNAs and messenger RNAs (mRNAs). We found an intriguing difference between miRNAs and siRNAs in their membrane-cytosol partitioning. We showed that all miRNAs, including 22-nt miRNAs, were associated with membrane-bound polysomes (MBPs) as opposed to polysomes in general. The plant miRNA effector AGO1 associated with membranes in part in an RNA-independent manner, recruited miRNAs to membranes, and exerted its endonuclease activity on MBPs. Reduced membrane-association of 22-nt miRNAs in an *ago1* mutant led to reduced levels, or loss of phasing, of phasiRNAs. The *TAS* transcripts, presumed to be noncoding, were actually bound by MBPs in a manner that supports phasiRNA production. These findings establish MBPs as the site of action of plant miRNAs and uncover a role of ribosomes and the ER in siRNA biogenesis, thus expanding the known functions of the rough ER.

## Results

### Differential partitioning of miRNAs and siRNAs between membranes and cytosol

We previously uncovered an integral ER membrane protein (named AMP1) as required for miRNA-mediated translational repression of target RNAs and showed that a few examined miRNAs and their target transcripts were present on MBPs (*Li et al., 2013*). To gain a global view of the membrane association of cellular small RNAs (sRNAs), we performed sRNA sequencing (sRNA-seq) for total (T) and microsomal (M; *Figure 1A*) RNAs. Three biological replicates were highly reproducible (*Figure 1—figure supplement 1A*). For an initial analysis, we quantified sRNAs by normalizing against total reads. The T sRNA population was characterized by two prominent size classes, 21 nt and 24 nt, as previously observed (*Kasschau et al., 2007*; *Lu et al., 2006*) (*Figure 1—figure*

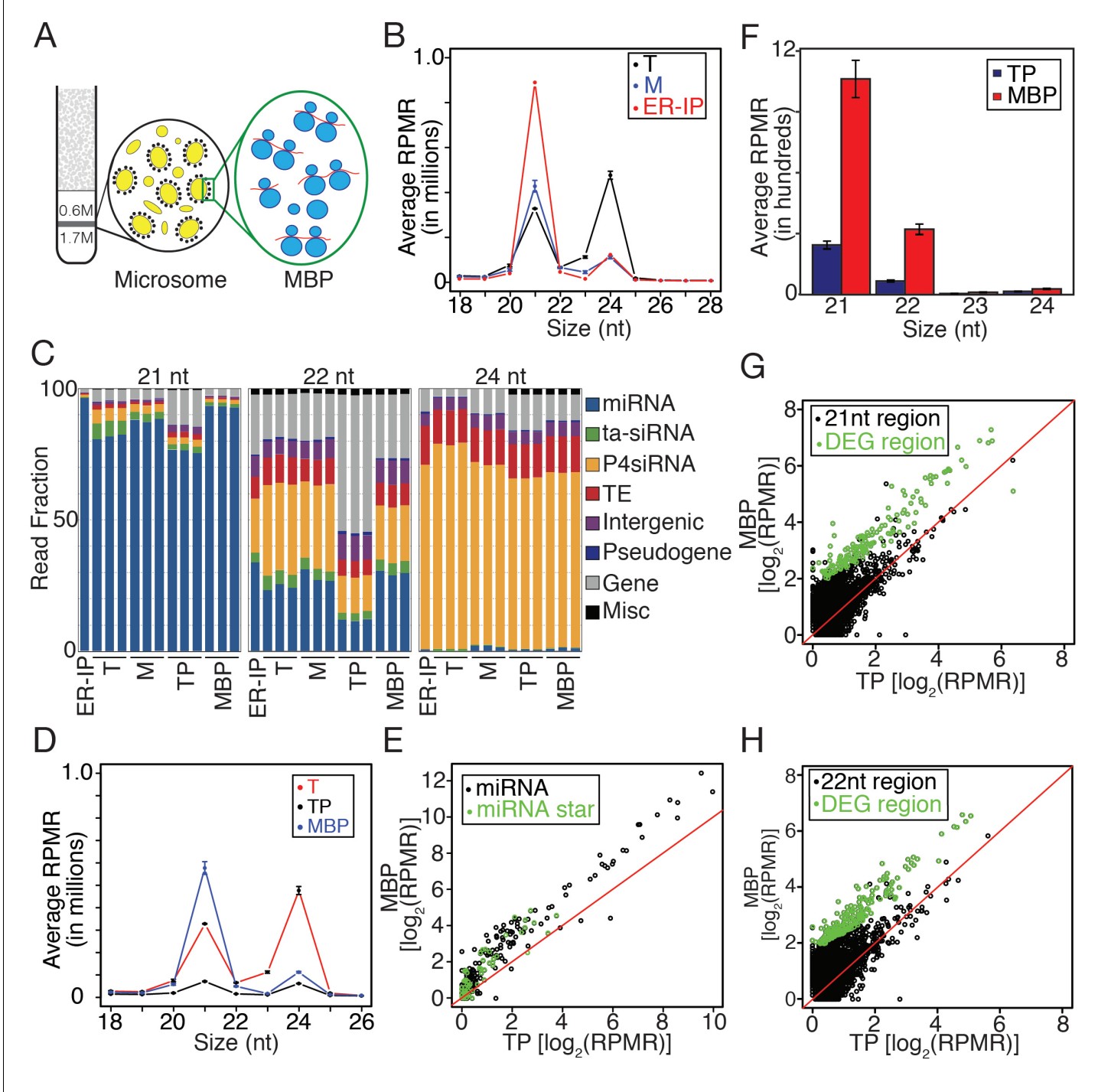

**Figure 1.** Enrichment of miRNAs and a set of endogenous siRNAs in the microsomal (M) and membrane-bound polysome (MBP) fractions. (**A**) A brief scheme of microsome and MBP isolation. (**B**) Size (in nucleotides) distribution of sRNAs in total extract (T), microsome (M), and an ER pull-down (ER-IP). RPMR, reads per million 45S rRNA (see Materials and methods). Error bar indicates standard deviation from three biological replicates. (**C**) Composition of sRNAs in three size classes in various samples. TP, total polysome; MBP, membrane-bound polysome. Each column represents a biological replicate. (**D**) Size (in nucleotides) distribution of sRNAs in T, TP, and MBP. (**E**) Abundance of annotated miRNAs and miRNA*s in MBP and TP. (**F**) Abundance of siRNAs from *TAS1*, *2*, *3*, and four loci in TP and MBP. All siRNAs mapping to the *TAS* loci were sampled in the four size classes. (**G–H**) A set of endogenous siRNAs was enriched in the MBP fraction. The genome was tiled into 100 bp windows and the abundance of 21-nt (**G**) or 22-nt (**H**) sRNAs in these windows were compared between MBP and TP samples. Regions showing enrichment (DEG region) are in green.

*Figure 1 continued on next page*

*Figure 1 continued*

The following figure supplements are available for figure 1:

**Figure supplement 1.** Analyses of sRNA-seq libraries.
**Figure supplement 2.** Comparison of MBP and TP sRNAs.

*supplement 1B*). In contrast, the M sRNA population exhibited a larger 21-nt peak and a diminished 24-nt peak (*Figure 1—figure supplement 1B*). In both T and M sRNA populations, miRNAs were the major component in the 21-nt class while P4siRNAs (Pol IV-dependent siRNAs) were a major component of the 24-nt class (*Figure 1—figure supplement 1C*). Notably, in the M fraction, the proportion of miRNAs was increased in 21-, 22-, and 24-nt classes (*Figure 1—figure supplement 1C*).

As the M fraction was a crude membrane preparation, and our previous study implicated the rough ER as the site of miRNA-mediated translational repression (*Li et al., 2013*), we sought to determine whether ER-associated sRNAs show a similar profile as M sRNAs. We took advantage of a transgenic line expressing an ER membrane protein SEC12 fused to YFP (YFP-SEC12) (*Agee et al., 2010*) to enrich for the ER. YFP-SEC12 was present in the M fraction as expected (*Figure 1—figure supplement 1D*), and the YFP tag was on the cytosolic side of the ER (*Figure 1—figure supplement 1E*). We pulled down ER using anti-YFP antibodies; western blotting showed that this ER preparation contained two ER proteins YFP-SEC12 and HSC70 but not ARF1 and SYP22, which marks Golgi and endosome, respectively (*Ebine et al., 2008*; *Ritzenthaler et al., 2002*) (*Figure 1—figure supplement 1F*). Similar to the M fraction, the ER preparation showed a skewed sRNA size distribution towards 21 nt (*Figure 1—figure supplement 1B*).

Because the M and T samples differed drastically in the composition of small RNAs, normalization against total reads were not appropriate if comparisons of sRNA levels between the two sample types were to be made. To establish a reasonable normalization method for such comparisons, we utilized published sRNA-seq libraries from wild type and the *nrpd1-1* mutant (*Li et al., 2015*). *NRPD1* encodes the largest subunit of Pol IV (*Herr et al., 2005*); the sRNA population in *nrpd1-1* lacks most 24-nt P4siRNAs (*Mosher et al., 2008*; *Zhang et al., 2007*) (*Figure 1—figure supplement 1H*), which resembles the M sRNA profile (*Figure 1—figure supplement 1B*). Although miRNA abundance is unaffected in Pol IV mutants (*Herr et al., 2005*; *Kanno et al., 2005*; *Onodera et al., 2005*), miRNAs appeared to be greatly increased in abundance when read counts were normalized against total mapped reads (*Figure 1—figure supplement 1I*), and this was also true for 21-nt sRNAs (*Figure 1—figure supplement 1H*). We found that if normalization was performed against 18–41-nt rRNA fragments (from 5.8S, 18S, and 25S rRNAs from 80S ribosomes), then the levels of 21-nt sRNAs and miRNAs were similar in WT and *nrpd1-1* (*Figure 1—figure supplement 1J,K*), which better reflected the actual situation. Thus, rRNA fragments could serve as an internal control in sRNA-seq quantification for our purposes. We refer to the normalized read counts as RPMR (reads per million rRNA fragments).

We re-analyzed T and M sRNA-seq using this normalization method, which was also used for all subsequent sRNA-seq analyses in this study. The size distribution of M sRNAs still showed a diminished 24-nt peak and an increased 21-nt peak (*Figure 1B*), indicating not only a depletion of 24-nt sRNAs but also an increase in 21-nt sRNAs in the M fraction. The M fraction had a higher proportion of miRNAs for 21-, 22-, and 24-nt size classes than the T samples (*Figure 1C*), indicating that miRNAs were enriched in the membrane fraction relative to other small RNAs.

The reduction in the 24-nt peak in the M fraction was due to a depletion of P4siRNAs. The *Arabidopsis* genome was divided into consecutive and non-overlapping 100 bp windows, and normalized sRNA read counts in each window were compared between M and T samples. P4siRNA read counts were much lower in the M fraction (*Figure 1—figure supplement 1G*). Note that the great majority of P4siRNAs reside in the cytoplasm, despite their nuclear functions in guiding DNA methylation (*Ye et al., 2012*). Therefore, most cytoplasmic P4siRNAs were not associated with membranes. Thus, this work revealed an intriguing difference in the partitioning between membranes and cytosol for miRNAs and P4siRNAs.

## Association of miRNAs with membrane-bound polysomes (MBPs)

Having shown that miRNAs were enriched in the M fraction relative to P4siRNAs and other siRNAs, we next asked whether they were present on MBPs. We isolated MBPs (diagrammed in *Figure 1A* and in more detail in *Figure 1—figure supplement 2A*; a representative MBP profile is shown in *Figure 1—figure supplement 2A*) in three biological replicates and performed sRNA-seq (reproducibility is shown in *Figure 1—figure supplement 1A*). MBP-associated sRNAs had a profile with a large 21-nt peak and a small 24-nt peak (*Figure 1D*). The proportions of miRNAs in all size classes were higher in MBP than in T (*Figure 1C*), suggesting that miRNAs were enriched in MBP relative to other sRNAs.

AGO1 co-fractionates with polysomes (*Lanet et al., 2009*), implying that miRNAs associate with polysomes. Thus, the observed MBP enrichment of miRNAs could be attributed to the association of miRNAs with polysomes in general. To determine whether miRNAs associated with MBPs, a more meaningful comparison would be between MBPs and total cellular polysomes (TPs). We isolated TPs (diagrammed in *Figure 1—figure supplement 2B*) and MBPs from the same seedling samples in three biological replicates and sequenced the associated sRNAs (reproducibility is shown in *Figure 1—figure supplement 1A*). The MBP samples had a much larger 21-nt peak relative to TP samples (*Figure 1D*), indicating that the enrichment of 21-nt sRNAs in MBP was due to their association with MBPs but not polysomes in general. For each size class, and particularly for the 22-nt class, the proportion of miRNAs was greatly increased in MBP relative to TP (*Figure 1C*).

The abundance of individual miRNAs in MBP and TP fractions was determined using reads that mapped exactly to annotated miRNAs or miRNA*s. Most miRNAs as well as some detectable miRNA*s showed higher abundance in MBP vs. TP (*Figure 1E*). 21-nt and 22-nt ta-siRNAs (from *TAS1* to *TAS4* loci) also had higher levels in MBP (*Figure 1F*). To evaluate the MBP enrichment of miRNAs and ta-siRNAs relative to other 21-nt or 22-nt sRNAs, we calculated 21-nt or 22-nt sRNA counts in 100 bp windows of the genome. Endogenous siRNAs constituted most of the 21-nt or 22-nt sRNA-generating regions of the genome. Only a small set of the windows produced MBP-enriched sRNAs (green circles in *Figure 1G,H*), suggesting that a small set of endogenous siRNAs was enriched on MBPs while the majority of siRNAs were not.

To identify MBP-enriched sRNAs, we compared the abundance of sRNAs in each 100 bp window for each sRNA size class (21, 22, 23, and 24 nt) between MBP and TP samples. This analysis revealed higher numbers of hyper (MBP>TP) than hypo (MBP<TP) differential sRNA regions (DSRs) (*Figure 1—figure supplement 2C*, *Supplementary file 1*). At the 21-nt and 22-nt hyper DSRs, 25–50% overlapped with *MIR* and *TAS* genes (*Figure 1—figure supplement 2D*).

The large number of 22-nt hyper DSRs was particularly intriguing, as 22-nt sRNAs have the unique capacity to trigger the biogenesis of phasiRNAs from their target genes. Among the 188 22-nt hyper DSRs, 31 overlapped with *MIR* genes representing 24 miR families (*Supplementary file 1*). This was surprising, as only a few miRNAs are annotated as 22 nt long. This prompted us to interrogate the sizes of all *Arabidopsis* miRNAs using our sRNA-seq datasets.

Among a total of 427 *Arabidopsis* miRNAs, 178 were at an average level > 1 RPMR in the 12 sRNA-seq samples (three samples each for T, M, MBP and TP from wild type) (*Supplementary file 2*). Of these 178 miRNAs, 141 and 17 were annotated as 21 nt and 22 nt, respectively (*Supplementary file 2*). By examining the actual sizes of the miRNAs, we found that many miRNAs had both 21-nt and 22-nt isoforms, and the annotated size represented the size of the major isoform (*Figure 2A*). 22-nt miRNAs, regardless of whether they were the major or minor isoforms, were more abundant in MBP relative to TP (*Figure 2B*).

We investigated how the 22-nt miRNA isoforms from predominantly 21-nt miRNA-producing loci arose. The 22-nt miRNA isoforms tended to have a genome-matched 1-nt extension relative to the 21-nt major isoforms, and 3' extension was more prevalent than 5' extension (*Figure 2C*). DCL1 is the Dicer that generates miRNAs, most of which are 21 nt long (*Park et al., 2002*; *Reinhart et al., 2002*), and DCL2 produces 22-nt siRNAs (*Gasciolli et al., 2005*). The abundance of 22-nt miRNA isoforms was greatly reduced in a *dcl1* mutant but not in a *dcl2* mutant (*Figure 2D*) (*Kasschau et al., 2007*), suggesting that the 22-nt miRNA isoforms were made by DCL1. Thus, precursor processing by DCL1 is not precise such that 22-nt miRNA isoforms from many *MIR* genes are made. This poses a potential problem, as 22-nt miRNA isoforms have the potential to trigger phasiRNA production from target transcripts.

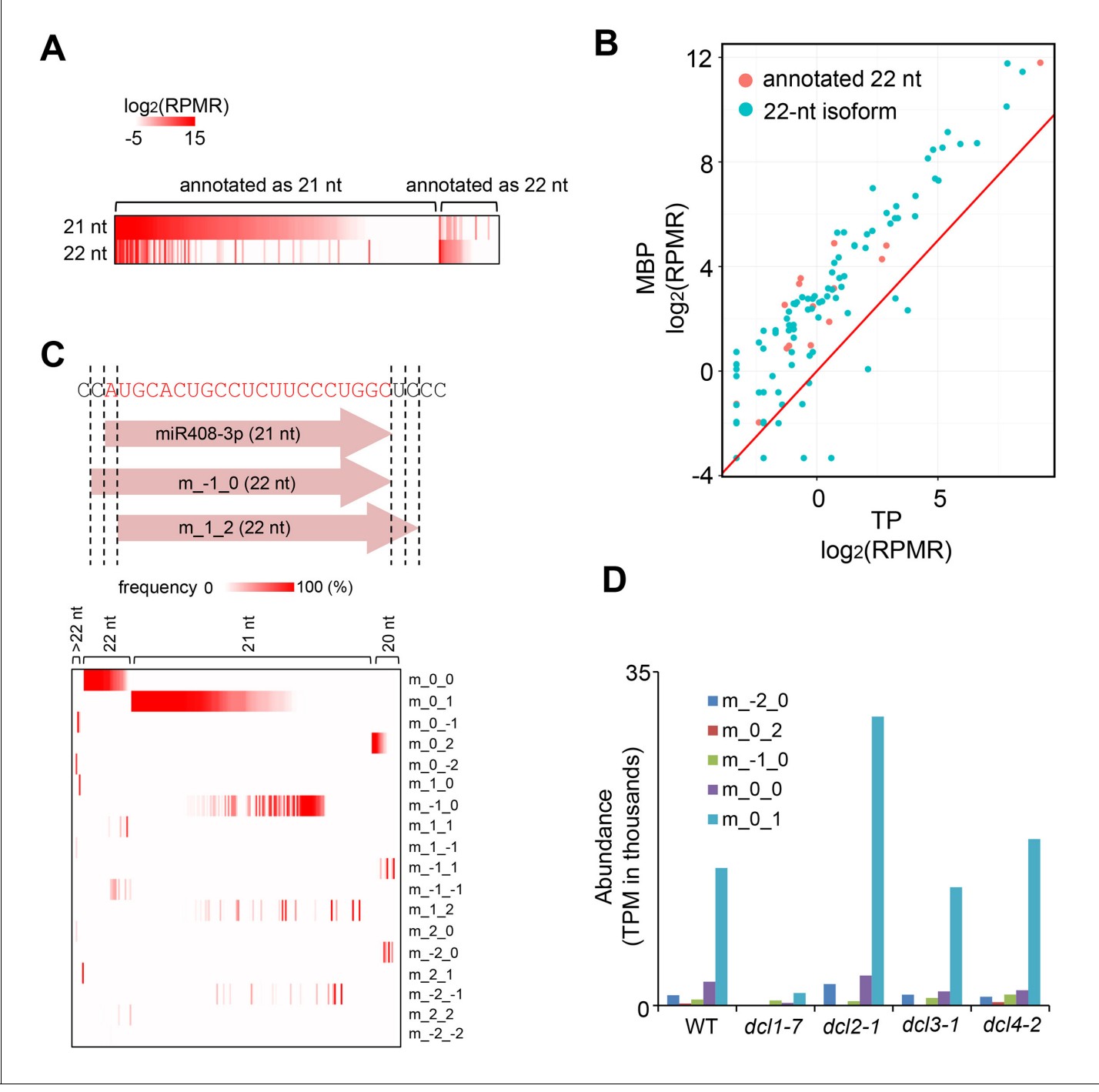

**Figure 2.** The production and MBP association of 22-nt miRNA isoforms. (**A**) Many miRNAs exist as both 21-nt and 22-nt isoforms. A heatmap showing the abundance of 21-nt and 22-nt isoforms of 178 miRNAs that were at levels > 1 RPMR in the 12 sRNA-seq samples (three samples each for T, M, MBP and TP from wild type). Each horizontal line represents a miRNA. The annotated sizes are indicated on top. The actual sizes are indicated to the left. (**B**) 22-nt miRNAs are enriched in MBP relative to TP. The abundance of 22-nt miRNAs in MBP and TP samples is shown. Only a few miRNAs (red dots) were from loci annotated to produce 22-nt miRNAs; most were 22-nt isoforms (cyan dots) from loci that produce predominantly 21-nt miRNAs. (**C**) Origins of 22-nt miRNA isoforms. The positions of the 5′ and 3′ ends of the 22-nt isoforms are defined relative to the annotated miRNA 5′ and 3′ ends using a scheme shown in the diagram of miR408-3p. This miRNA exists predominantly as a 21-nt form. 'm_-1_0' represents a 22-nt isoform with an extra nucleotide on the 5′ end. 'm_1_2' represents a 22-nt isoform that is shifted relative to the 21-nt isoform by 1-nt and 2-nt at the 5′ and 3′ ends, respectively. A heatmap showing the abundance of 22-nt isoforms that fall into the various categories. The annotated sizes of the miRNAs are shown on the top. The various mechanisms to generate a 22-nt isoform are indicated to the right. 3′ extension is the predominant mechanism that generates 22-

*Figure 2 continued on next page*

*Figure 2 continued*

nt isoforms from loci that predominantly produce 21-nt or 20-nt miRNAs. (D) DCL1 generates 22-nt miRNA isoforms. The abundance of 22-nt miRNA isoforms was determined using published sRNA-seq from wild type (WT) and various *dcl* mutants (GSE6682) (*Fahlgren et al., 2007*). The various mechanisms of 22-nt miRNA production are as defined in (C). TPM, transcript per million.

## The MBP-associated transcriptome

Given that miRNAs were enriched in the MBP fraction, we asked whether they could meet their target transcripts on MBPs. The canonical view is that transcripts encoding proteins that enter the secretary pathway are co-translationally recruited to the ER. Under such a scenario, many miRNA target transcripts, which encode transcription factors, are not expected to be on MBPs. However, we found that a few miRNA target transcripts were present on MBPs (*Li et al., 2013*). To gain a global view of MBP-associated mRNAs, we performed polyA+ RNA-seq of MBP RNAs and total RNAs. We found that the overall profiles of MBP and total RNAs were highly similar (data not shown), suggesting that all cellular mRNAs were represented on MBPs.

To further validate this, we switched to another method of fractionation that allowed for the simultaneous recovery of MBPs and cytosolic free polysomes (FPs) (*Figure 3—figure supplement 1*). Note that this purification scheme could not result in MBPs as pure as the previous method (*Figure 1—figure supplement 2A*), but was used here to allow the comparison between MBP and FP RNAs. In brief, total extracts were first separated into microsome and cytosol fractions, from which MBPs and FPs were then isolated, respectively. The MBP and FP profiles on sucrose gradients were as expected (*Figure 3—figure supplement 1A*). The microsome, cytosol, MBP, and FP RNAs were then subjected to polyA+ RNA-seq in three biological replicates (reproducibility is shown in *Figure 3—figure supplement 1B*). Clustering analysis using the 5000 top varying transcripts (*Supplementary file 3.1*) showed that the microsome and MBP fractions clustered together and the cytosol and FP fractions clustered together (*Figure 3A*). This indicated that the separation of these fractions was successful. We next examined transcripts encoding transmembrane domain-containing proteins; these transcripts were expected to be translated on MBPs (*Supplementary file 3.2*). Indeed, many of these transcripts showed higher abundance in microsome than cytosol (green circles in *Figure 3B*) and in MBP than FP (green circles in *Figure 3C*). This further validated the successful fractionation. Most cellular transcripts were similarly represented on MBPs and FPs, with a large fraction over-accumulated on MBPs (*Figure 3D*). Likewise, the majority of predicted and known targets of miRNAs (*Supplementary file 3.3*) were equally represented on MBPs and FPs, with few being depleted from MBPs (*Figure 3E*). This suggested that nearly all cellular transcripts are translated on MBPs as well as FPs.

## RNA-independent membrane association of AGO1

AGO1 partially co-localizes with ER and co-fractionates with microsomes and MBPs (*Brodersen et al., 2012*; *Li et al., 2013*). We examined M and MBP fractions for the presence of AGO1 and AGO4 by western blotting. As expected, AGO1 was found in both M and MBP fractions (*Figure 4A*). AGO4, which binds most 24-nt P4siRNAs (*Qi et al., 2006*), was not detected in either fraction (*Figure 4A*), which is consistent with the observed depletion of P4siRNAs from these fractions (*Figure 1—figure supplement 1G*).

To determine whether AGO1's membrane association relied on miRNAs, we isolated microsomes from mutants in miRNA biogenesis genes *DCL1* and *HYL1* (*Han et al., 2004*; *Park et al., 2002*; *Reinhart et al., 2002*; *Vazquez et al., 2004a*). Microsomal AGO1 levels were unaffected in these mutants (*Figure 4B*; *Figure 4—figure supplement 1*), suggesting that AGO1's membrane association was most likely miRNA-independent. To determine whether AGO1's membrane association relied on long RNAs, we treated total cell extracts with RNase I followed by microsome isolation. Full-length *CSD2* RNA (a target of miR398) was eliminated by RNase I treatment (*Figure 4C*). AGO1, miR398, and miR156 were still present in the M fraction (*Figure 4C,D*), although their levels were reduced. Therefore, AGO1's membrane association was largely independent of intact target RNAs.

To investigate whether AGO1's MBP association was RNA-dependent, we first isolated MBPs and then fractionated MBPs by sucrose gradient centrifugation to separate the light fraction containing

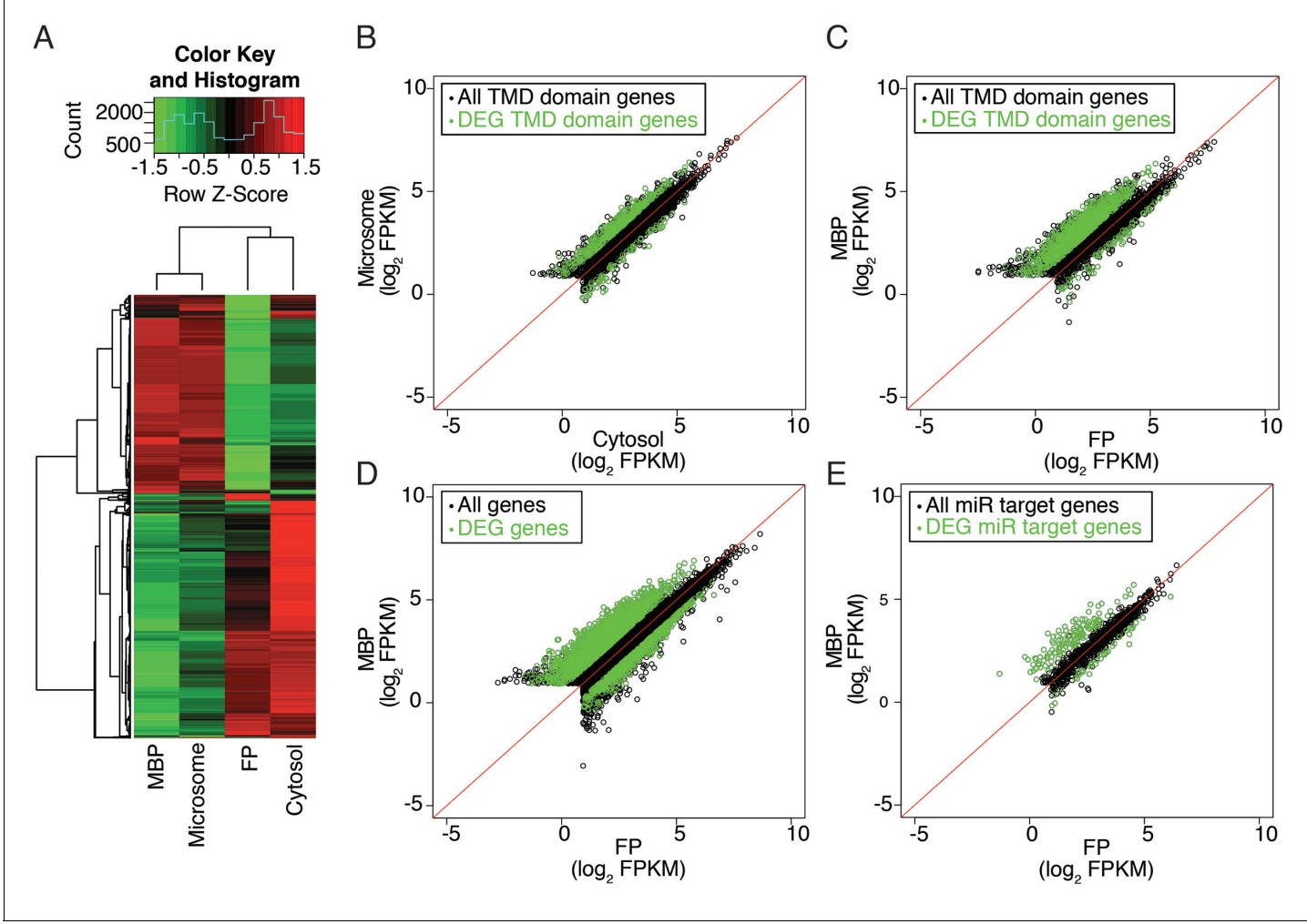

**Figure 3.** Distribution of cellular mRNAs in microsome versus cytosol and MBP versus FP. RNA-seq was performed in three biological replicates from polyA+ RNA isolated from microsome, cytosol, MBP and FP. (A) Clustering analysis of the four sample types with the 5000 top varying transcripts. (B–E) partitioning of various groups of transcripts between microsome and cytosol or MBP and FP. FPKM, Fragments Per Kilobase of transcript per Million mapped reads. Enriched and depleted transcripts (DEG transcripts; FC>2; FDR < 0.05) are highlighted in green. (B–C) Abundance of transcripts encoding transmembrane domain (TMD) proteins in microsome vs. cytosol (B) and MBP vs. FP (C). (D–E) The partitioning of all transcripts (D) or miRNA target transcripts (E) between MBP and FP. The miRNA target transcripts were predicted with psRNATarget (*Dai and Zhao, 2011*) with the maximum expectation score ≤ 3.

The following figure supplement is available for figure 3:

**Figure supplement 1.** Simultaneous isolation of microsome, cytosol, free polysome (FP) and MBP followed by RNA-seq.

ribosomal subunits and 80S monosomes from the polysomes (*Figure 4E*; right panel). AGO1 was found in both the light fraction and the polysomes (*Figure 4E*; bottom panel). We digested MBPs with RNase I; the sucrose gradient profiles of the digested MBPs showed that the polysomes were reduced to monosomes (*Figure 4E*; left panel). AGO1 was still detectable in this monosome fraction but at much reduced levels (*Figure 4E*; bottom panel). This showed that AGO1' MBP-association depended to a large extent on intact mRNAs.

## AGO1-dependent membrane association of miRNAs

Having shown that AGO1's membrane association was largely independent of miRNAs or mRNAs, we next sought to determine whether miRNAs' membrane association depended on AGO1. If

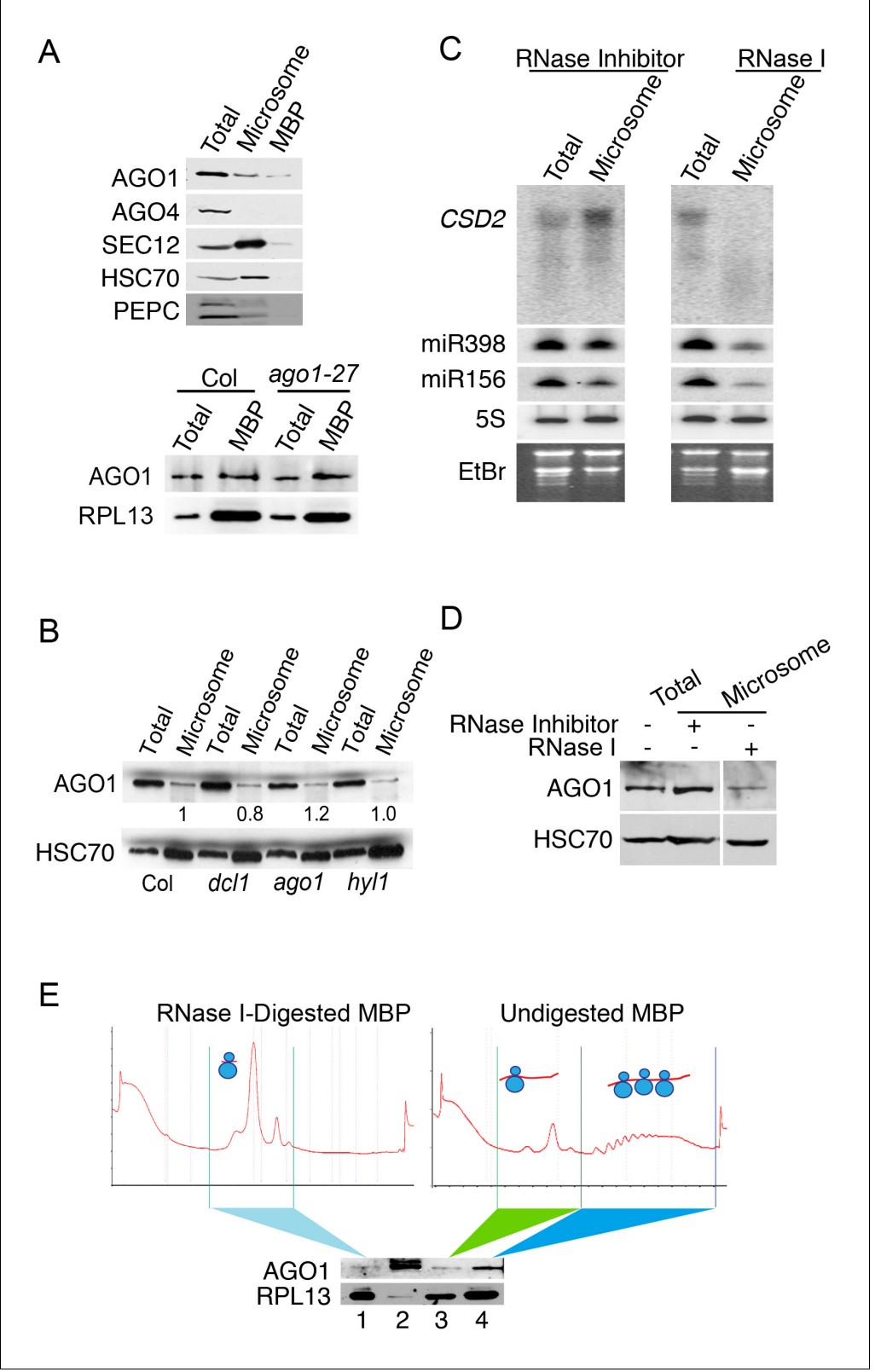

**Figure 4.** The microsome- and MBP-association of AGO1. (**A**) Western blot to detect various proteins in total extract and microsomal and MBP fractions. SEC12 is an ER transmembrane protein; HSC70 is an ER luminal protein; PEPC is a cytosolic protein; RPL13 is a ribosomal protein. AGO1 but not AGO4 was present at detectable levels in microsomal and MBP fractions. The bottom panel shows that the AGO1-27 protein was still associated

*Figure 4 continued on next page*

*Figure 4 continued*

with MBPs. (B) Microsomal AGO1 and HSC70 levels as determined by western blotting in wild type (Col), *dcl1-20* (*dcl1*), *ago1-27* (*ago1*), and *hyl1-2* (*hyl1*). Note that *dcl1-20* is a newly *isolated*, strong *dcl1* allele (see *Figure 4—figure supplement 1*). (C–D) Microsomal miRNA (C) and AGO1 (D) levels with or without RNase I treatment. Microsomes were isolated from extracts that were treated with RNase inhibitor or RNase I and subjected to northern blotting to detect *CSD2* mRNA and miRNAs and western blotting to detect the AGO1 protein. 5S rRNA and the stained gels (EtBr) indicate relative sample loading. (E) The association of AGO1 with monosomes and polysomes. MBPs were treated or not with RNase I and fractionated in a sucrose gradient. The marked fractions were subjected to western blotting to detect AGO1 and the ribosomal protein L13 (RPL13). Lane 2, input (total extract). Note that for 'undigested MBP', the polysome fraction is expected to have more ribosomes than the monosome fraction. Thus, the higher RPL13 levels in the polysome fraction do not reflect unequal loading. On the other hand, RPL13 levels in the 'RNase I-digested MBP' can be compared to those of the polysomal 'Undigested MBP' to reflect relative sample loading.

The following figure supplement is available for figure 4:

**Figure supplement 1.** Characterization of a new *dcl1* allele.

---

AGO1 recruits miRNAs to the ER, then the levels of microsomal miRNAs should be reduced in *ago1* mutants. We first examined the *ago1-36* mutant that lacks a full-length AGO1 protein and exhibits severe morphological defects (*Baumberger and Baulcombe, 2005*) (*Figure 5A*). Despite a global reduction in miRNA accumulation in this mutant (*Vaucheret et al., 2004*), a few miRNAs accumulated to levels similar to wild type (*Figure 5A*), presumably because these miRNAs associated with other AGOs or non-AGO proteins. The levels of these miRNAs in microsomes were reduced in *ago1-36* (*Figure 5A*), suggesting that AGO1 promoted the membrane association of these miRNAs.

In order to obtain a global view of the membrane association of miRNAs in an *ago* mutant, we resorted to the weaker *ago1-27* allele from which sufficient plant material could be obtained. Three biological replicates of sRNA-seq were performed for total extracts (T) and two for the microsomal fraction (M) in this mutant (*Figure 1—figure supplement 1A*). Consistent with previous observations (*Vaucheret et al., 2004*), the *ago1-27* mutation did not have a strong overall effect on miRNA accumulation (*Figure 5B*). The membrane association of miRNAs, represented by the M/T ratio, was drastically reduced in *ago1-27* for most miRNAs (*Figure 5C*), suggesting that the membrane association of miRNAs depended on AGO1. In this analysis, miR390 was an internal control - it is bound by AGO7 (*Montgomery et al., 2008*) and thus its membrane association should not be affected by the *ago1-27* mutation. The M/T ratio of miR390 was largely unchanged in *ago1-27* (*Figure 5C*). The M/T ratio was also reduced for most 21–22-nt ta-siRNAs (from *TAS1* to *TAS4* loci) in *ago1-27* (*Figure 5D*). As ta-siRNAs are bound by AGO1 (*Baumberger and Baulcombe, 2005*), this also supports the conclusion that AGO1 recruits sRNAs to membranes.

## miRNA-guided cleavage in M and MBP fractions

The subcellular location of miRNA-guided target RNA cleavage has been unknown and perhaps presumed to be in the cytosol. We investigated whether miRNA-guided cleavage was detectable in the M fraction. AGO1 was immunopecipitated from total extracts and from microsomes, and slicer assay was carried out by incubating AGO1 immunoprecipitates with radiolabeled *PHB* RNA containing the miR165/6-binding site. This RNA was cleaved by AGO1 immunoprecipitates from both total extracts and microsomes (*Figure 6A*).

3' cleavage fragments can be detected in vivo by 5' RACE RT-PCR (*Llave et al., 2002*). We searched for 3' cleavage fragments from M and MBP RNAs by 5' RACE RT-PCR. Cloning and sequencing of the 3' cleavage fragments from a few miRNA target genes showed that cleavage occurred precisely at the expected positions for both M and MBP RNAs (*Figure 6B,C*).

## phasiRNA production from miRNA target transcripts on MBPs

The finding of miRNA-guided cleavage occurring on MBPs prompted us to study the relationship between the MBP association and the phasiRNA-triggering activity of miRNAs. We focused on small RNAs known to cause phasiRNA production from their target transcripts. These were miR161.1

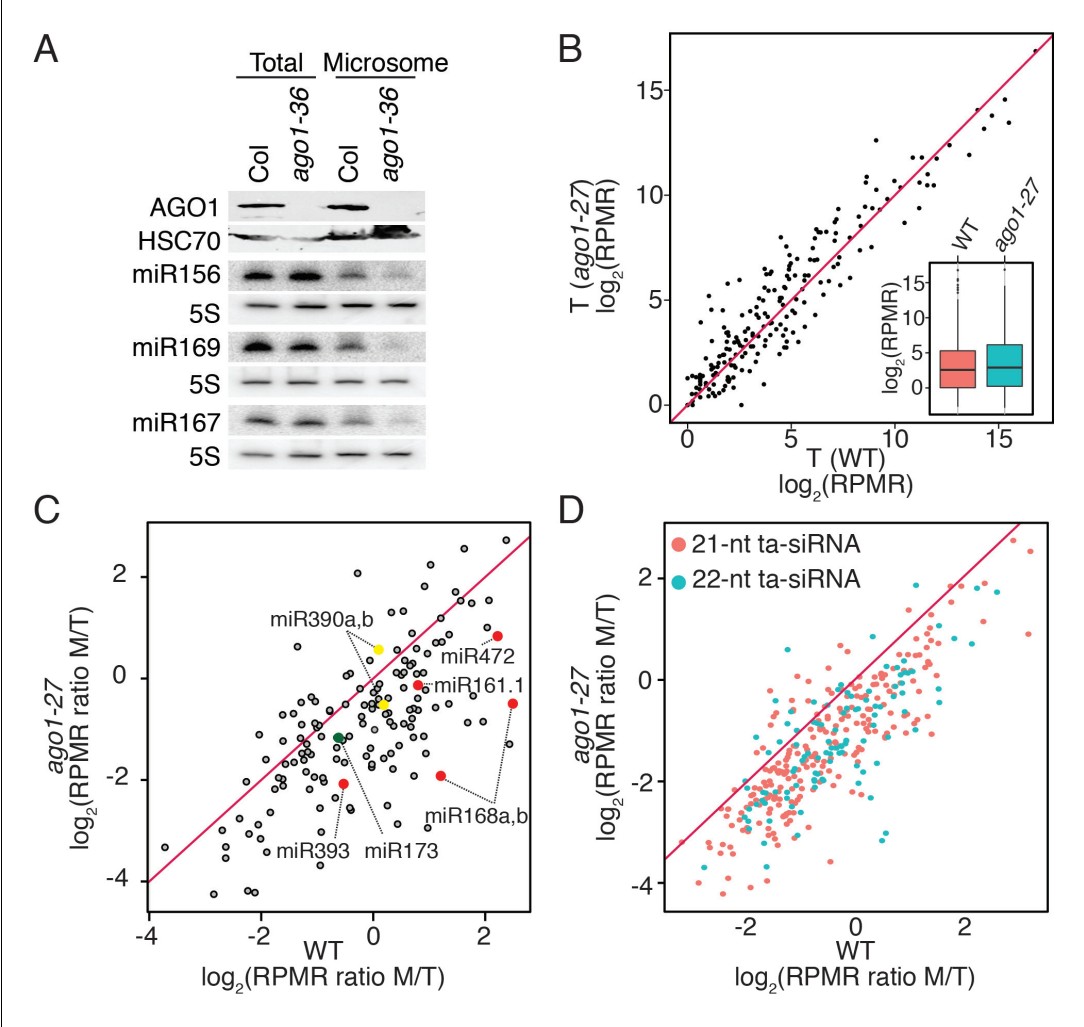

**Figure 5.** miRNAs and ta-siRNAs are recruited to membranes by AGO1. (**A**) Detection of three miRNAs in total extracts and the microsomal fraction in wild type (Col) and *ago1-36*. The *ago1-36* mutant lacks the full-length AGO1 protein as shown by western blotting. HSC70 was a loading control. The three miRNAs were detected by northern blotting. 5S rRNA was an internal control and shown below each miRNA blot. The microsomal levels of the miRNAs were reduced in *ago1-36*. (**B**) A scatter plot showing comparable miRNA abundance in wild type (WT) and *ago1-27* total extracts. Inset: box plots illustrating the distribution of miRNA RPMR values in WT and *ago1-27*. (**C**) A scatter plot showing that miRNAs have reduced microsomal enrichment in *ago1-27* as compared with wild type. Microsomal enrichment was measured by M/T (ratio of levels in microsome vs. those in total extract). The degree of microsomal enrichment of miR390 (yellow dots) is largely unchanged, while that of most known ta-siRNA/phasiRNA triggers (red dots) is decreased in *ago1-27*. miR173's (green dot) microsomal enrichment was weakly affected. (**D**) A scatter plot showing that 21-nt and 22-nt ta-siRNAs from *TAS1-4* loci have reduced microsomal enrichment in *ago1-27* as compared with wild type.

targeting *PPR* genes, miR168 targeting *AGO1*, miR173 targeting *TAS1* and *TAS2*, miR393 targeting auxin receptor genes (*AFB2* and *AFB3*), miR472 and miR825* targeting *NBS-LRR* genes, and miR828 targeting *TAS4*. Among these miRNAs, miR173, miR393 and miR828 are predominantly 22 nt long while the others are predominantly 21 nt long with 22-nt isoforms. We also identified two new pha-siRNA-generating loci: *CIL2* (At3g23690) encoding a basic helix-loop-helix protein and being targeted by miR393, and At5g43740 encoding an NBS-LRR protein and being targeted by miR472 (*Figure 7—figure supplement 1A*). As a control, we also included miR390, which is 21 nt long, bound by AGO7, and triggers ta-siRNA production from *TAS3* loci in a 'two hit' mode. In addition, ta-siR2140 is a ta-siRNA from *TAS2*; this ta-siRNA is 22 nt long and able to trigger phasiRNA production from a *PPR* gene (*Chen et al., 2007*).

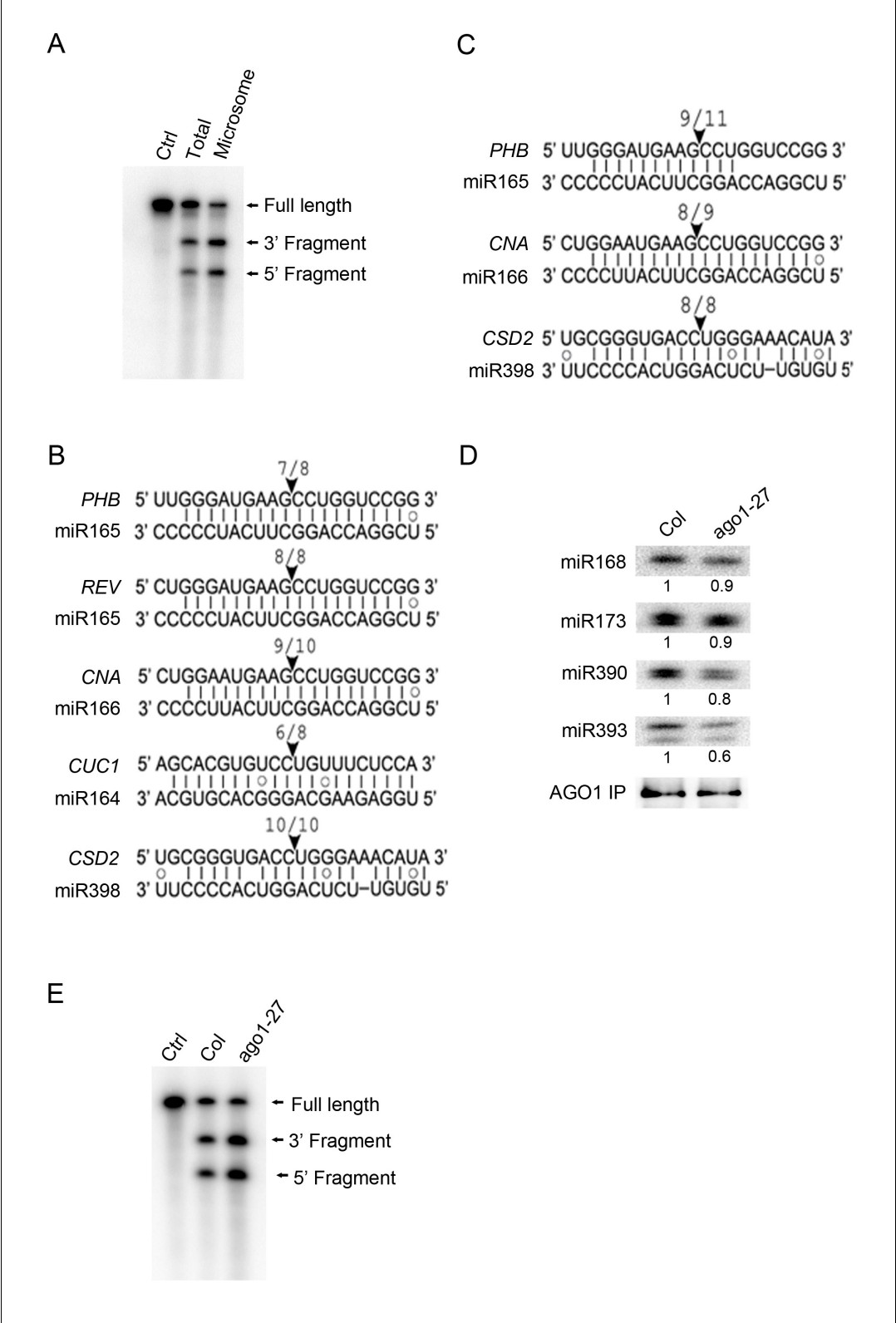

**Figure 6.** miRNA-guided cleavage occurs in the microsomal and MBP fractions and in *ago1-27*. (**A**) In vitro slicer assay using AGO1 immunoprecipitates (IP) from total extracts or the microsomal fraction. A fragment of *PHB* RNA containing the miR165/6-binding site was used as the substrate (marked 'Full Length'). The two cleavage products are indicated. The control (Ctrl) lane was the RNA alone without AGO1 IP. (**B–C**) Detection of 3' cleavage fragments from various miRNA target RNAs in vivo using 5' RACE RT-PCR from the microsomal (**B**) or MBP (**C**) fraction. The cloned PCR products were

*Figure 6 continued on next page*

*Figure 6 continued*

sequenced to identify the 5' ends of the 3' cleavage fragments. The arrowheads indicate the positions of miRNA-guided cleavage. The numbers above indicate the number of clones with 5' ends at the predicted cleavage site out of total sequenced clones. (D) The AGO1-27 protein associated with miRNAs in vivo. IP was performed in wild type and *ago1-27* and the IP was subjected to western blotting to detect AGO1 and northern blotting to detect several miRNAs. The levels of miRNAs were quantified against the levels of AGO1 in the IP and compared between wild type and *ago1-27*. (E) In vitro slicer assay using AGO1 IP from wild type and *ago1-27* total extracts. The assay was performed as in (A).

The levels of these sRNAs were unaffected or mildly reduced in *ago1-27* in total extracts (*Figure 7A*; top panel). However, the abundance of these sRNAs in microsomes was much lower in *ago1-27*, indicating that these sRNAs associated with membranes in an AGO1-dependent manner (*Figure 7A*; top panel). As expected, neither the overall nor the microsomal abundance of miR390 was reduced in *ago1-27* (*Figure 7A*; lower panel).

Levels of phasiRNAs were determined by sRNA-seq in *ago1-27*, *ago1-36*, and wild type. As expected, phasiRNAs from all 23 examined loci (*Supplementary file 4*; *TAS3* loci excluded) were

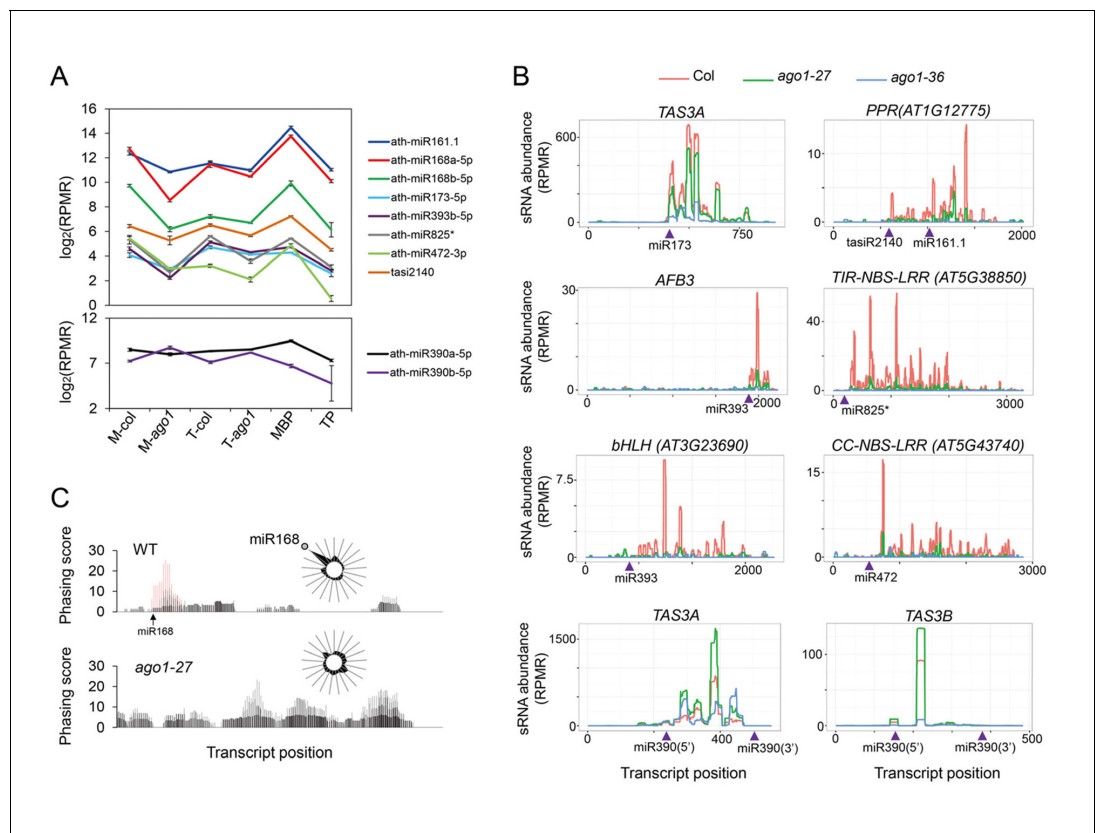

**Figure 7.** Biogenesis of phasiRNAs is defective in *ago1-27*. (A) top panel: abundance of known AGO1-dependent ta-siRNA/phasiRNA triggers in microsome (M), total (T), MBP and TP as determined by sRNA-seq; lower panel: abundance of miR390 in M, T, MBP and TP as determined by sRNA-seq. Wild type: col; *ago1-27*: ago1. Error bars indicate standard error of the mean (n = 3). (B) sRNAs from known ta-siRNA/phasiRNA loci, except for *TAS3* loci, were reduced in *ago1-27* (green) as compared with wild type (red), and nearly diminished in *ago1-36* (blue). The loci IDs are shown above each plot. The miRNA triggers for these ta-siRNAs/phasiRNAs are shown below each plot with the miRNA-binding sites indicated by triangles. (C) The phasing of sRNAs over the *AGO1* transcript in wild type and *ago1-27*. miR168 triggers the production of phasiRNAs (with positions marked by red lines) in wild type, and the phasing caused by this miRNA was nearly lost in *ago1-27*. The radial graphs show that the miR168 cleavage site fell in the most prominent phasing register of 21-nt sRNAs in wild type but not in *ago1-27*.

The following figure supplement is available for figure 7:

**Figure supplement 1.** The effect of *ago1-27* on ta-siRNAs/phasiRNAs.

nearly eliminated in *ago1-36* (*Figure 7B*, *Figure 7—figure supplement 1B*). For 21 of the 23 pha-siRNA loci examined, the overall abundance of siRNAs was significantly reduced in *ago1-27* (*Figure 7B*, *Figure 7—figure supplement 1B*, *Supplementary file 4*). In *ago1-27*, the siRNAs from *AGO1* RNA were similar in abundance to wild type, but the phasing triggered by miR168-guided cleavage was nearly lost (*Figure 7C*). Notably, miR168 was one of the most affected miRNAs in terms of membrane association in *ago1-27* (*Figure 5C*). As a control, the abundance of *TAS3* ta-siRNAs (triggered by miR390-AGO7) was not reduced (they were in fact increased) in *ago1-27* (*Figure 7B*). The phasing of *TAS3* ta-siRNAs was unaffected (*Figure 7—figure supplement 1C*). Therefore, the reduction in phasiRNAs levels from non-*TAS3* loci in *ago1-27* was not due to inability of the AGO1-27 protein to bind phasiRNAs. The AGO1-27 protein was still associated with miRNAs, as shown by AGO1 IP followed by northern blotting to detect miRNAs (*Figure 6D*). The AGO1-27 protein was still capable of miRNA-guided cleavage in an in vitro slicer assay (*Figure 6E*), consistent with the previous conclusion that the *ago1-27* allele is normal in miRNA-guided cleavage in vivo (*Brodersen et al., 2008*). Thus, reduced phasiRNA abundance, or loss of phasing, coincided with decreased membrane association of the triggering miRNAs in *ago1-27*. These data, together with the finding that miRNA-guided cleavage occurs in M and MBP fractions, suggested that phasiRNA production occurs, or at least starts, on MBPs. This is also consistent with the reported membrane association of ta-siRNA biogenesis factors (*Jouannet et al., 2012*).

If phasiRNA biogenesis occurs on MBPs, we expect to detect the above-mentioned miRNA target transcripts on MBPs. We performed ribosome-protected fragment sequencing (ribo-seq) with MBPs. In brief, MBPs were first isolated (*Figure 1—figure supplement 2A*) and then treated with RNase I to resolve polysomes to monosomes. The monosomes were recovered from sucrose gradient centrifugation (as in *Figure 4E*), and monosome-protected RNA fragments were subjected to high-throughput sequencing. Regular RNA-seq was also performed with MBP RNAs. Our ribo-seq was successful as reflected by the fact that reads were derived predominantly from coding regions and regions close to the start and stop codons in UTRs, whereas MBP RNA-seq reads covered entire transcripts (*Figure 8A*). All above-mentioned, phasiRNA-generating, protein-coding genes were indeed present on MBPs (*Figure 8B* and data not shown).

Most importantly, we found that the *TAS* transcripts, which were thought to be non-protein-coding, were associated with MBPs. Although the entire *TAS* transcripts were associated with MBPs, as revealed by RNA-seq of MBP RNAs (*Figure 8C*), there was a sharp boundary between ribosome-protected and non-protected portions of each *TAS* transcript (*Figure 8C*). The ribosome-protected portions corresponded to the short open reading frames (ORFs) present in these RNAs. Amazingly, the miRNA-binding site was at the boundary of ribosome-protected and non-protected portions in each *TAS* transcript (*Figure 8C*), and the region unbound by ribosomes generated ta-siRNAs (compare sRNA-seq reads with ribo-seq reads in *Figure 8C*). An earlier study showed that the miRNA-binding site needs to be close to the upstream stop codon for phasiRNA biogenesis from an artificial phasiRNA-generating locus (*Zhang et al., 2012*). These observations suggest that the initial triggering step (i.e. miRNA-guided cleavage) in phasiRNA biogenesis occurs on MBPs and ribosome occupancy plays a role in specifying which transcript or portion of a transcript undergoes phasiRNA biogenesis.

## Discussion

Previously, we found that the integral ER membrane protein AMP1 is required for the translational repression activity of plant miRNAs, suggesting that the rough ER is the site of miRNA-mediated translational repression (*Li et al., 2013*). A few miRNAs and their target transcripts were found to be associated with MBPs (*Li et al., 2013*), but the scale of the MBP-association of miRNAs and their target transcripts was unknown. In this study, we examined the ER- and rough ER-association of small RNAs as well as cellular mRNAs. We found that most cellular mRNAs were enriched on MBPs or equally partitioned between MBPs and FPs. This indicates that most mRNAs, including miRNA target transcripts, are translated both on the rough ER and in the cytosol. AGO1 was previously shown to associate with polysomes, thus implicating the presence of miRNAs on polysomes. In this study, we found that miRNAs are associated with MBPs rather than polysomes in general.

AMP1 is only required for the translational repression activity of plant miRNAs, thus it was not known whether miRNA-guided cleavage occurs on MBPs. In this study, we found that miRNA-guided

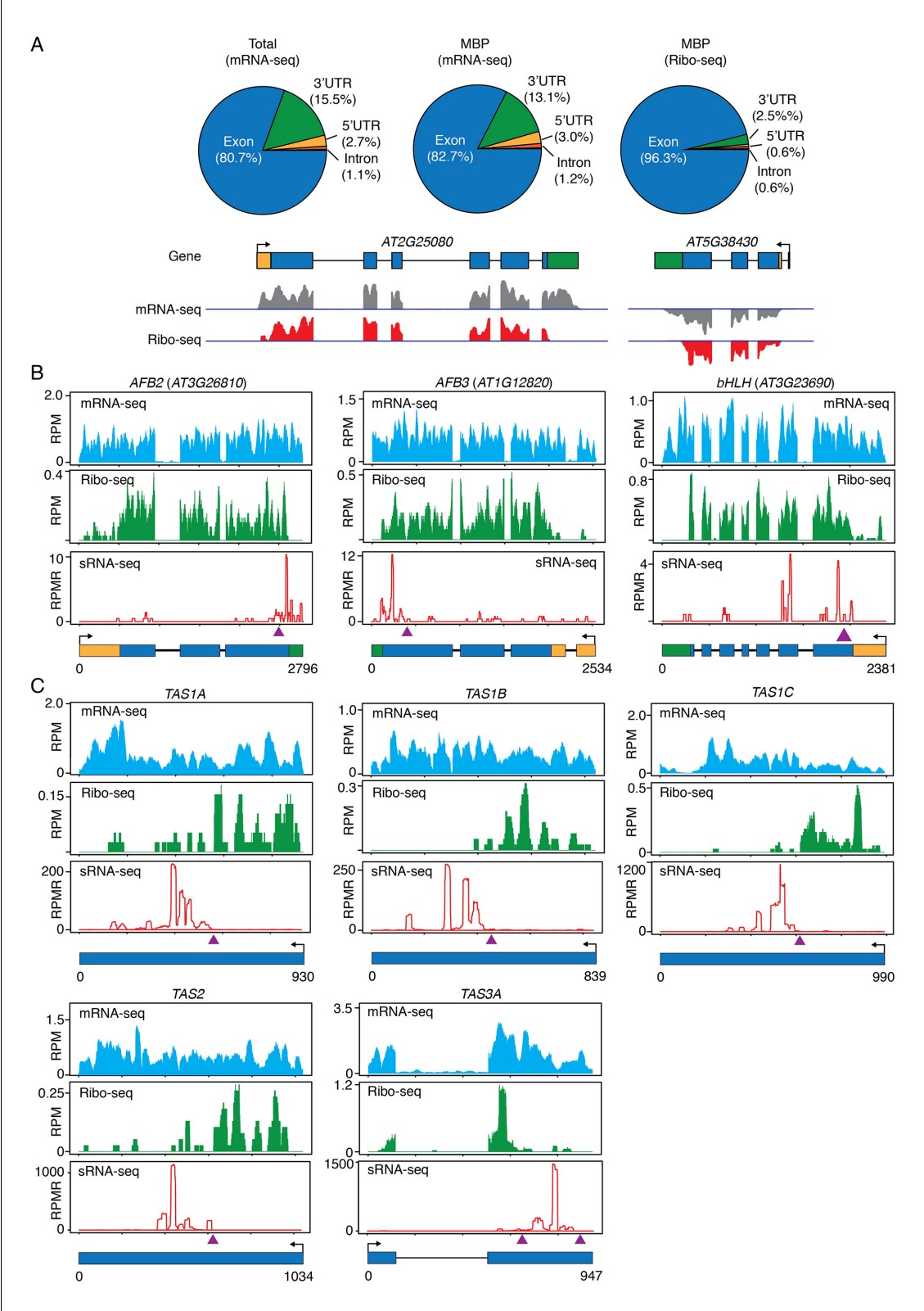

**Figure 8.** Ribo-seq to examine ribosome occupancy of transcripts known to spawn phasiRNAs. (**A**) Composition of the genomic features represented by reads from mRNA-seq from total extract (Total), mRNA-seq from MBP, and ribo-seq from MBP. The overwhelming representation of exons by ribo-seq shows that ribo-seq successfully captured the expected sequences. Two genes are shown as examples. In the gene models, rectangles and lines represent exons and introns, respectively. 5' and 3' UTRs are in yellow and green, respectively. Arrows mark the direction of transcription. The lack of

*Figure 8 continued on next page*

*Figure 8 continued*

most 3' UTR reads is consistent with the fact that ribosomes only protect a small portion of the 3' UTR. (**B–C**) Browser views of MBP mRNA-seq (top panels), ribo-seq (middle panels), and sRNA-seq (lower panels) for three protein-coding genes (**B**) and five *TAS* loci (**C**). The gene IDs are shown above the plots. The gene models are shown below the plots. The same scheme is used to represent exons, introns, UTRs, and transcription start sites as in (**A**). The triangles represent the positions of the binding sites of the triggering miRNAs.

cleavage does occur on MBPs. However, it should be noted that our findings do not exclude the occurrence of miRNA-guided cleavage in the cytosol. Our studies with the null allele *ago1-36* and the weak allele *ago1-27* showed that AGO1 is instrumental in recruiting miRNAs to the ER. But it is not known how the weak *ago1-27* mutant compromises the membrane association of miRNAs, as the mutant protein was still membrane- and MBP-associated (*Figure 4A–B*).

We found that phasiRNA production from most known phasiRNA-generating loci was reduced in *ago1-27*, in which the triggering miRNAs showed reduced membrane association. The *TAS* transcripts, despite not having long open reading frames, were present on MBPs and were bound by ribosomes in a manner that 'exposed' the ta-siRNA-generating portions. These findings indicate that phasiRNA production occurs, or is at least initiated, on MBPs. This is consistent with the finding that phasiRNA biogenesis factors reside in siRNA bodies that are in close proximity to a cis-Golgi marker (*Jouannet et al., 2012*). A recent study showed that *TAS3* RNA is bound by ribosomes, and AGO7 causes ribosome stalling upstream of the miR390 binding site (*Hou et al., 2016*). Our study confirms the ribosome association of *TAS3* RNA, extends the observation of ribosome occupancy to other *TAS* RNAs, and reveals the association of *TAS* RNAs with MBPs.

However, we do not believe that MBPs offer an optimal environment for phasiRNA production. We found that many *MIR* genes produce 22-nt isoforms due to imprecision in DCL1 processing, yet most 22-nt miRNA isoforms do not trigger phasiRNA production from their target RNAs. We propose that the sequestration of 22-nt miRNAs and miRNA target transcripts on MBPs prevents them from engaging in phasiRNA production. We propose that the rough ER is somehow capable of inhibiting the entry of most protein-coding transcripts into the phasiRNA biogenesis pathway. The *TAS* genes and a handful of protein-coding genes have evolutionarily adapted to the rough ER environment by having an optimal arrangement between the miRNA binding site and ribosome occupancy to enable phasiRNA biogenesis. Thus, ER and ribosomes have a previously unknown function in regulating the biogenesis of small RNAs.

## Materials and methods

### Plant materials

*Arabidopsis thaliana* (RRID:SCR_004618) wild-type (Columbia accession), *ago1-27* (*Vaucheret et al., 2004*), *hyl1-2* (*Han et al., 2004*; *Vazquez et al., 2004a*), and *SUC2:amiR-SUL* (*de Felippes et al., 2011*) strains were used in this study.

### Microsome and MBP isolation

The isolation of microsomes and MBPs for small RNA sequencing was performed as described (*Li et al., 2013*). In brief, 2 g of seedlings were ground in 7 ml microsome isolation buffer (MEB: 100 mM Tris-HCl, pH7.5, 5 mM EGTA, 15 mM MgCl$_2$, 5 mM DTT, 0.3M sucrose, 50 U/ml superaseIN (Invitrogen, Thermo Fisher Scientific, Waltham, MA, USA), 50 µg/ml cycloheximide, 50 µg/ml chloramphenicol, and a complete proteinase inhibitor cocktail (Roche)), and the cell lysate was filtered by 2 layers of miracloth and centrifuged at 10,000 g twice to remove debris. A 100 µl aliquot was taken from the clarified supernatant as input, and the rest was applied onto a sucrose step gradient (2.5 ml 20%/2.5 ml 60% sucrose), and centrifuged at 100,000 g in an SW41 rotor (Beckman coulter) for 1 hr. Microsomes were recovered from the interface of the two sucrose layers, diluted with 10 volumes of MEB and precipitated by centrifugation at 100,000 g for 30 min.

To isolate MBP, the microsome preparation was lysed with 8 ml polysome isolation buffer (0.2M Tris-HCl, pH9.0, 0.2M KCl, 0.025M EGTA, 0.035M MgCl2, 0.2% Brij-35, 0.2% Triton X-100, 0.2% Igepal CA630, 0.2% Tween 20, 0.2% polyoxyethylene 10 tridecyl ether, 5 mM DTT, 1 mM PMSF, 50 µg/

ml cycloheximide, 50 µg/ml chloramphenicol and 2.5 U/ml superaseIN). The lysate was kept on ice for 30 min, and then loaded on the top of a sucrose cushion (0.4M Tris-HCL, pH9.0, 0.2M KCl, 0.005M EGTA, 0.035m MgCl$_2$, 1.75M sucrose, 5 mM DTT, 50 µg/ml cycloheximide, and 50 µg/ml chloramphenicol) and centrifuged at 170,000 g for 3 hr. The pellet was dissolved with resuspension buffer (0.2M Tris-HCl, pH9.0, 0.2M KCl, 0.025M EGTA, 0.035M MgCl$_2$, 5 mM DTT, 50 µg/ml cyclo-heximide, and 50 µg/ml chloramphenicol) and the solution was transferred into a new tube as the MBP fraction. An aliquot of the MBP was subjected to 15–50% sucrose gradient centrifugation for polysome integrity evaluation, and the rest was used for sRNA and mRNA sequencing.

## Isolation of cytosol, microsome, free polysomes (FPs) and membrane-bound polysomes (MBPs)

A detailed protocol for the isolation of cytosol, microsome, free polysomes (FP) and membrane-bound polysomes (MBP) may be found at Bio-protocol (*Zhao and Li, 2017*). The simultaneous recovery of cytosol and microsomal fractions from the same samples, followed by the isolation of FPs and MBPs from the two fractions, respectively, was performed according to (*de Jong et al., 2006*). Briefly, 2 g of seedlings were ground in liquid nitrogen, and the powder was resuspended with 8 ml extraction buffer (0.2M Tris-HCl, pH8.5, 0.1M KCl, 70 mM Mg(C$_2$H$_3$O$_2$)$_2$, 50 mM EGTA, 0.25M sucrose, 10 mM DTT, 50 ug/ml cycloheximide, 50 ug/ml chloramphenicol and 2.5 U/ml superaseIN). The slurry was filtered by 2 layers of miracloth and centrifuged twice at 10,000 g for 10 min to remove debris. Microsomes were pelleted by centrifugation at 30,000 g for 30 min in a Beckman SW28 rotor. The supernatant was transferred into a new tube as the cytosol fraction, and the micro-some pellet was dissolved with 8 ml extraction buffer supplemented with 1% v/v Triton X-100. Both the cytosol and microsomal fractions were centrifuged again at 30,000 g for 30 min to remove any residual membranes, and the supernatant was loaded onto an 8 ml 1.75M sucrose cushion as described in the 'Microsome and MBP isolation' section and centrifuged at 170,000 g for 3 hr. The FP and MBP pellets were resuspended with 400 µl resuspension buffer as described in the 'Micro-some and MBP isolation' section, and polysome integrity evaluation was conducted as before.

Note that this procedure does not recover MBPs that are as pure as the procedure described above, but allows the simultaneous recovery of both cytosol and microsomes. This MBP isolation approach was only used when FP and MBP transcriptomes were compared. RNA was extracted from the cytosol and microsomal fractions by TRI-reagent (Molecular Research Center) and subjected to RNA-seq (see below).

## Isolation of total polysomes (TPs)

Total cellular polysomes were isolated as previously described (*Mustroph et al., 2009*). Briefly, 1g seedlings were ground in liquid nitrogen and the powder was resuspended in 8 ml polysome extrac-tion buffer (see the 'Microsome and MBP isolation' section). The slurry was clarified by centrifugation at 10,000 g for 10 min at 4°C twice. The supernatant was loaded onto an 8 ml 1.75M sucrose cush-ion, and ribosomes were pelleted at 170,000 g for 3 hr at 4°C in a Type70 Ti rotor (Beckman Coulter) and resuspended in 400 µl resuspension buffer (see the 'Microsome and MBP isolation' section) for RNA isolation and analysis of polysome profiles.

## Determination of AGO1 levels in microsomal and MBP fractions

1.5 g seedlings were ground in liquid nitrogen and the powder was resuspended in 4 ml MEB buffer. The slurry was clarified by passage through two layers of miracloth and centrifugation at 10,000 g for 10 min twice at 4°C. The supernatant was subjected to further centrifugation at 100,000 g for 1 hr at 4°C to separate the cytosol (the soluble) and microsomal (the pellet) fractions. The pellet was washed by the same amount of MEB once again to avoid cytosol contamination of the microsomal fraction.

AGO1 and AGO4 were detected by western blotting with anti-AGO1 or anti-AGO4 antibodies (AGO1: AgriSera Cat# AS09 527 RRID:AB_2224930; AGO4: AgriSera Cat# AS09 617 RRID:AB_ 10507623). To test the RNA dependence for AGO1's membrane association, the clarified plant extract was treated with either RNase inhibitor (as a control) or 80 units/ml RNase I at room temper-ature for 1 hr, after which microsomes were isolated and subjected to RNA and protein analyses as described above.

To detect AGO1's association with MBPs, MBPs were isolated as described in the 'Microsome and MBP isolation' section, and subjected to sucrose gradient centrifugation to separate the light fraction containing the 40S, 60S, and 80S ribosomes from the polysomes. Proteins were isolated from these fractions using TRI reagent (Molecular Research Center) following manufacturer's instructions. To determine the RNA-dependence of AGO1's MBP association, 1000 O.D.260 units of MBPs were treated with 1500 units of RNase I followed by sucrose gradient centrifugation. The light fraction containing the 40S, 60S, and 80S ribosomes and the heavy fraction containing polysomes were recovered. Proteins were isolated from these fractions with TRI reagent following manufacturer's instructions and subjected to western blotting to detect AGO1 and the ribosomal protein L13 with commercial antibodies (AgriSera Cat# AS09 478 RRID:AB_2060757).

## ER immunoprecipitation

1 g seedlings expressing YFP-SEC12 were ground in liquid nitrogen into fine powder, and the powder was resuspended with 2 ml IP buffer containing 50 mM Tris-HCl (pH7.5), 150 mM NaCl, 0.3M sucrose, 10% glycerol, and 1x proteinase inhibitor (Roche). 100 µl supernatant was saved as the input after clarification by 10,000 g centrifugation at 4°C, and the rest was used for immunoprecipitation. Briefly, the extract was pre-cleared by Dynabeads (Invitrogen), incubated with anti-GFP antibodies (Clontech Laboratories, Inc. Cat# 632592 RRID:AB_2336883) for 2 hr at 4°C with gentle rotation, and washed by IP buffer four times. The beads were boiled with 1x protein loading buffer, and the proteins were subjected to western blotting to detect YFP-SEC12, HSC70, ARF1, SYP22, and PEPC using commercial antibodies (anti-GFP: Clontech Laboratories, Inc. Cat# 632380 RRID:AB_10013427; anti-HSC70: Enzo Life Sciences Cat# ADI-SPA-818-F RRID:AB_11180122; anti-ARF1: AgriSera Cat# AS08 325 RRID:AB_1271007; anti-SYP22: a gift from Dr. Natasha Raikhel; PEPC: AgriSera Cat# AS09 458 RRID:AB_2063166). Using the same procedure, ER IP was performed followed by RNA isolation and sRNA sequencing (see below for sRNA library construction and data analysis).

## Membrane topology analysis for the YFP-SEC12 protein

Microsomes from YFP-SEC12 transgenic plants were isolated, resuspended with MEB buffer, and digested with 10 ng/µl proteinase K (Fermentas) in the presence or absence of 1% Triton X-100 for 20 min on ice. The reactions were stopped by the addition of phenylmethylsulfonyl fluoride to a final concentration of 5 mM and subjected to western blotting using anti-GFP antibodies (Clontech Laboratories, Inc. Cat# 632380 RRID:AB_10013427) to detect YFP-SEC12.

## AGO1 immunoprecipitation (IP) and slicer assay

To IP AGO1 from total cell extracts, the same procedure as that of YFP-SEC12 IP was applied except that AGO1 antibody (AgriSera Cat# AS09 527 RRID:AB_2224930) was used. To IP AGO1 from microsomes, microsomes were first isolated and dissolved in IP buffer containing 1% of Triton X-100. The microsome lysate was then diluted by 10 volumes of non-detergent-containing IP buffer to reduce the Triton X-100 concentration to 0.1%, and immunoprecipitation was performed. After washes, the beads were resuspended in the reaction buffer for slicer activity assay (see below).

For the slicer activity assay, part of the *PHB* gene spanning the miR165/6 binding site was amplified with primers PHB-T7-F and PHB-R (*Supplementary file 5*). In vitro transcription was carried out with T7 RNA polymerase (Promega) in the presence of $^{32}$P UTP. The reaction mix was resolved on 5% polyacrylamide/7M urea gel. The band corresponding to the expected *PHB* transcript was excised from the gel, and the RNA was recovered by soaking the gel slicer in 0.3M NaCl overnight followed by precipitation with ethanol. The beads containing AGO1 immunoprecipitates were mixed with the *PHB* transcript in reaction buffer (1 mM ATP, 0.2 mM GTP, 1.2 mM MgCl$_2$, 25 mM creatine phosphate, 30 mg/mL creatine kinase and 0.4 unit/mL RNase Inhibitor (Promega)). The reaction mix was incubated at room temperature for 2 hr. RNA was then extracted with TRI-reagent and resolved in a 5% polyacrylamide/7M urea gel, and $^{32}$P signals were detected with a Phosphoimager.

## Detection of 3' cleavage products in vivo

To detect the 3' cleavage products from miRNA targets, 5' RACE was performed using the GeneRacer kit (Invitrogen). Total RNAs were directly ligated with the GeneRacer RNA oligonucleotide and reverse transcription reactions were carried out with an oligo dT primer. Seminal quantitative PCR

was performed with gene specific primers (*Supplementary file 5*) to quantify 3' cleavage products from miRNA targets. To map the cleavage position, the PCR products were cloned into the T-easy vector (Promega) and subjected to Sanger sequencing.

## Small RNA library construction

30 µg of RNA was resolved on 15% urea-PAGE gel, and the 18–30-nt region was excised from the gel. Small RNAs were recovered by soaking the smashed gel in 0.3M NaCl overnight, followed by precipitation with ethanol. Small RNA libraries were constructed following instructions from the Illumina Truseq small RNA library preparation kit (Illumina). The libraries were sequenced on an Illumina Hiseq 2500 at the UC Riverside Genomics core facility.

## RNA-seq library construction

RNA-seq libraries were constructed using the NEBNext mRNA Library Prep kit (NEB). Briefly, polyA + RNAs were isolated with the Dynabeads mRNA DIRECT Purification Kit (ThermoFisher Scientific), and fragmented to approximately 200 nt. cDNA was synthesized with random primers, and DNA adaptors were ligated to both ends of the double-stranded cDNAs. PCR was performed with primers complementary to the sequences in the DNA adaptors to generate RNA-seq libraries, which were sequenced on Illumina Hiseq2500.

## Analysis of sRNA-seq

Reads from Illumina sRNA-seq were first processed to remove the 3' adaptor sequences (TGGAATTCT or AGATCGGAA) and then size-selected (18 to 42 nt) using cutadapt v1.9.1 (*Martin, 2011*). Adaptor-free reads that aligned to rRNA/tRNA/snoRNA regions were then removed using Bowtie v1.1.0 (*Langmead et al., 2009*) with the parameters '-v 2 k1', and reads that were assigned to the 45S rRNA regions were counted. The rest of the reads were mapped to the TAIR10 genome (http://www.arabidopsis.org) using ShortStack v3.4 (*Johnson et al., 2016*) with parameters ' –bowtie_m 1000 –ranmax 50 –mmap u –mismatches 0 '. To calculate and compare small RNA abundance in different genotypes or samples, the genome was tiled into 100 bp windows and reads whose 5' end nucleotides fall within a window were assigned to the window. Normalization was performed by calculating the RPMR value (<u>r</u>eads <u>p</u>er <u>m</u>illion of 45S <u>r</u>RNA reads) for each window, and comparison was conducted by the R package 'DESeq2' (*Love et al., 2014*).

Annotated miRNA sequences were obtained from miRBase v21. To quantify miRNA abundance, adaptor-free reads were searched for those that matched to the sequence of each miRNA allowing for a 1-nt shift on either the 5' or 3' end. To quantify ta-siRNA levels, reads located within each of the eight annotated *TAS* gene regions (*TAS1a*, *1b*, *1c*, *TAS2*, *TAS3a*, *3b*, *3c*, and *TAS4*) were counted and summed separately for each size class (21 nt, 22 nt, 23 nt, and 24 nt). Windows that harbor P4siRNAs were determined as described (*Li et al., 2015*).

## Small RNA phasing analysis

Phasing analysis was performed as previously described (*De Paoli et al., 2009*; *Howell et al., 2007*). Small RNA reads from sense and antisense strands were unified. Phasing score was calculated by the following formula.

$$\text{Phasing score} = \ln \left[ \left( 1 + 10 \times \frac{\sum_{i=1}^{10} \text{Pi}}{1 + \sum^{U}} \right)^{n-2} \right], n > 3$$

Where n = number of phase cycle positions occupied by $\geq$1 reads within a 10-cycle window. P = total number of reads for all sRNAs with start coordinates in a given phase within a 10-cycle window. U = total number of reads for all sRNAs with start coordinates outside of a given phase.

## Processing and mapping of mRNA-seq reads

Raw RNA-seq reads that passed the Illumina quality control steps were collapsed into a set of non-redundant reads. These non-redundant reads were mapped to the TAIR10 *Arabidopsis* genome using TopHat v2.0.4 with default settings (*Kim et al., 2013*). Reads that were mapped to multiple positions were excluded from further analyses. Cuffdiff (*Trapnell et al., 2012*) was used to calculate

normalized gene expression levels and call differentially expressed genes (with FDR < 0.05 and fold change > 2).

## Ribo-seq library construction

A total of 2000 O.D.260 units of MBP was digested by 3000 units of RNase I (Ambion) in 400 µl digestion buffer (20 mM Tris-HCl, pH 8.0, 140 mM KCl, 5 mM MgCl2, 50 µg/ml cycloheximide, 50 µg/ml chloramphenicol) with gentle shaking at room temperature for 1.5 hr, and the digestion product was centrifuged in a 15–50% sucrose density gradient at 300,000 g for 1.5 hr. A gradient fractionation system (ISCO) was used to collect the monosome fraction. Monosomal RNAs were extracted with TRI Reagent (MRC), and resolved on 15% Urea-PAGE. RNAs corresponding to 25–35 nt were recovered from the gel and phosphorylated by T4 polynucleotide kinase (NEB). Next, the monosomal RNAs were further polyadenylated with PolyA Polymerase (NEB), and reverse transcribed with the primer oligodT linkerM (*Supplementary file 5*), followed by cDNA circularization with CircLigase (EpiCenter). rRNAs were removed by hybridization with biotin conjugated DNA probes (*Supplementary file 5*) followed by capture with Dynabead M-280 Streptavidin (Invitrogen). Finally, the Ribo-seq library was obtained by PCR with primers listed in *Supplementary file 5*.

## Ribo-seq analysis

Ribo-seq and MBP mRNA-seq datasets were processed using Tophat 2.0 with default parameters (*Trapnell et al., 2012*). Files containing genomic coordinates (e.g., 5' UTR, exon, intron) were obtained from TAIR 10. Reads were associated with a genomic feature using bedtools v2.23.0 (*Quinlan and Hall, 2010*) with overlap ≥80%.

## DATA deposition

The genomics datasets were deposited at NCBI GEO under the accession number GSE82041.

## Acknowledgements

We thank Drs. Natasha Raikhel and Detlef Weigel for sharing antibodies and genetic material. We thank Drs. Shou-Wei Ding and Wenrong He for comments on the manuscript. The work was supported by grants from Gordon and Betty Moore Foundation (GBMF3046), NIH (GM061146), NSFC (91440105), and Guangdong Innovation Research Team Fund (2014ZT05S078) to X Chen, NSFC (31210103901) to X Cao and the science technology and innovation committee of Shenzhen municipality (JCYJ20151116155209176, KQCX2015033110464302) to S Li.

## Additional information

### Funding

| Funder | Grant reference number | Author |
|---|---|---|
| Shenzhen municipality | JCYJ20151116155209176 | Shengben Li |
| Shenzhen municipality | KQCX2015033110464302 | Shengben Li |
| National Natural Science Foundation of China | 31210103901 | Xiaofeng Cao |
| Howard Hughes Medical Institute | | Xuemei Chen |
| Gordon and Betty Moore Foundation | GBMF3046 | Xuemei Chen |
| National Institutes of Health | GM061146 | Xuemei Chen |
| Guangdong Innovation Research Team Funds | 2014ZT05S078 | Xuemei Chen |
| National Natural Science Foundation of China | 91440105 | Xuemei Chen |

The funders had no role in study design, data collection and interpretation, or the decision to submit the work for publication.

## Author contributions
SLi, Conception and design, Acquisition of data, Analysis and interpretation of data, Drafting or revising the article; BL, XM, SLi, Analysis and interpretation of data, Drafting or revising the article; CY, LG, Analysis and interpretation of data; YY, BZ, LL, TS, YZ, Acquisition of data; BM, XCa, Conception and design; XCh, Conception and design, Analysis and interpretation of data, Drafting or revising the article

## Author ORCIDs
Xuemei Chen, http://orcid.org/0000-0002-5209-1157

# Additional files

### Supplementary files
• Supplementary file 1. Lists of differential small RNA regions (DSR) in the MBP versus TP comparison.

• Supplementary file 2. A list of 178 miRNAs in 12 sRNA-seq libraries.

• Supplementary file 3. Transcript levels in microsome, cytosol, free polysome and membrane-bound polysome fractions as determined by RNA-seq.

• Supplementary file 4. Changes in the abundance of phasiRNAs from known phasiRNA-generating loci caused by *ago1-27* and *ago1-36* mutations.

• Supplementary file 5. Oligonucleotides used in this study.

### Major datasets
The following dataset was generated:

| Author(s) | Year | Dataset title | Dataset URL | Database, license, and accessibility information |
|---|---|---|---|---|
| Li S, Le B, Ma X, You C, Yu Y, Zhang B, Liu L, Gao L, Shi T, Zhao Y, Mo B, Cao X, Chen X | 2016 | Endoplasmic reticulum as a hub organizing the activities of microRNAs and a subset of endogenous siRNAs in Arabidopsis | https://www.ncbi.nlm.nih.gov/geo/query/acc.cgi?acc=GSE82041 | Publicly available at the NCBI Gene Expression Omnibus (accession no. GSE82041) |

The following previously published dataset was used:

| Author(s) | Year | Dataset title | Dataset URL | Database, license, and accessibility information |
|---|---|---|---|---|
| Kasschau KD, Fahlgren N, Chapman EJ, Sullivan CM, Cumbie JS, Carrington JC | 2007 | Arabidopsis thaliana small RNAs sequences identified using high-throughput 454 sequencing technology | https://www.ncbi.nlm.nih.gov/geo/query/acc.cgi?acc=GSE6682 | Publicly available at the NCBI Gene Expression Omnibus (accession no. GSE6682) |

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
