## [Decision Letter]

Thank you for submitting your article "Biogenesis of phased siRNAs on membrane-bound polysomes in Arabidopsis" for consideration by *eLife*. Your article has been reviewed by three peer reviewers, one of whom is a member of our Board of Reviewing Editors, and the evaluation has been overseen by Detlef Weigel as the Senior Editor. The following individual involved in review of your submission has agreed to reveal his identity: Gary Ruvkun (Reviewer #3).

The reviewers have discussed the reviews with one another and the Reviewing Editor has drafted this decision to help you prepare a revised submission.

Summary:

In this paper, you combined genomic approaches with cellular fractionation to investigate the global patterns of cytoplasmic partitioning of small RNAs and "long" RNAs. You show that miRNAs and their target RNAs are associated with membrane-bound polysomes (MBPs). Moreover, cleavage activity and products are detected in microsomes and MBPs. You propose that miRNA cleavage occurs on MBPs. You further show that phasiRNA-triggering 22-nt miRNAs and their targets including non-coding TAS transcripts are also present on MBPs and propose phasiRNA biogenesis is initiated on MBPs.

The reviewers agreed that such findings should in principle be published at the highest levels. While the reviewers furthermore agreed that the data presented are clear and thorough, they also agreed that a couple of additional experiments would be essential to improve the manuscript.

Essential revisions:

1) Use dcl1 mutants to test whether miRNAs are required for the membrane association of AGO1. Alternatively, examine whether small RNA binding-deficient AGO1 is associated with microsomes. Small RNA binding is required for AGO1 stability. It would be very interesting if AGO1 is still associated with microsomes in the absence of miRNAs. The authors should add some discussions about this.

2) Examine whether the mutant form of AGO1 in ago1-27 binds miRNAs by IP-sRNA northern blot and whether it is associated with microsomes or MBPs by western blot.

We have included the full reviews, which mention separate points for revisions. The experiments listed in points #3, 4 and 5 of Reviewer #1 are also nice, but after discussion, we agree that these experiments are not essential for this paper. Reviewer #2 had a few minor comments, we ask you to take them into account as well.

Reviewer #1:

In this paper, the authors examined patterns pf subcellular partitioning of small RNAs and "big" RNAs at the genomic scale, using cellular fractionation followed by RNA deep sequencing. They found that miRNAs and their targets were associated with membrane-bound polysomes (MBPs). They also found that the AGO1 complexes purified from microsomal fraction (M) had slicer activity and 3'cleavage products could be detected in M and MBP fractions. They thus conclude that MBPs are the places where miRNA-guided cleavage occurs. More interestingly, the authors found that the phasiRNA-triggering 22-nt miRNAs and their targets including non-coding TAS transcripts were present on MBPs, suggesting phasiRNA biogenesis may be initiated on MBPs. These findings provide new insights into miRNA-guided target cleavage and phasiRNA biogenesis. Detailed comments are listed below.

1) The authors used a hyl1 mutant to show that the association of AGO1 with M is mostly likely independent of miRNAs. The hyl1 mutations actually have relatively moderate effects on miRNA accumulation, compared to that caused by dcl1 mutations. I would suggest the authors use dcl1 mutant instead for this experiment. Alternatively, the authors could also examine whether sRNA binding deficient AGO1 associates with M. It would be nice to do the same experiment with MBPs.

2) The authors found that the association of miRNAs with M was greatly reduced in ago1-27 suggesting that membrane association of miRNAs depended on AGO1. As ago1-27 is a week allele and the mutation has no strong effects on miRNA accumulation and slicer activity, suggesting that the mutant AGO1-27 can still bind miRNAs. The authors can test this by AGO1 IP followed by small RNA Northern blot. This is confirmed. Then the next question is how the mutation affects miRNA association with M. Does this mutation affects AGO1 association with M? This can be easily addressed by an AGO1 Western blotting with M/MBP fractions in ago1-27.

3) The authors detected cleavage activity/products in M and MBP fractions. This only indicates that miRNA cleavage can occur on MBPs but does not exclude the possibility that cleavage can also occur in other places. It would be nice if the authors can try to detect cleavage activity/products in MBP-depleted fractions.

4) It is very interesting that ribosome occupancy may play a role in specifying a sequence for phasiRNA biogenesis. I would suggest the authors to examine the ribosome profiles of the transcripts targeted by 22-nt miRNAs that are not capable of triggering phasiRNA biogenesis.

5) It would be also interesting to examine whether the components in the phasiRNA biogenesis pathway including SGS3 and RDR6 are associated with M/MBP fractions.

Reviewer #2:

Post-transcriptional gene silencing can occur via transcript cleavage or inhibition of translation. Currently there is little known about the mechanism of translational inhibition in plants. This manuscript builds on previous work from the Chen lab that indicated the ER is required for microRNA-mediated translational repression. The manuscript showed differential distribution of sRNA classes in the cytosol compared with those associated with microsomes, and that microRNAs were enriched in membrane-bound polysome preparations. This membrane association in part required AGO1. Interestingly, the ago1-27 mutant, which retains AGO1 cleavage activity, showed loss of phasing of miR168-mediated target cleavage. Furthermore, and unexpectedly, TAS transcripts were found to be associated with ribosomes. Ribosome-bound regions were not uniformly distributed across these transcripts and appeared to be confined to regions upstream of the miRNA binding site. These results hint at possible new roles for membrane-bound ribosomes as a control point in sRNA production. Overall the data are novel, interesting, clear and thorough. I have only a few minor comments on the manuscript.

Figure 1—figure supplement 1 F: there is no reference to PEPC.

Figure 4. What is the sample in lane 2?

Figure 6: Is it possible that residual cytosolic AGO1 in the microsome fraction is responsible for the observed target mRNA cleavage?

There are two [Supplementary-material SD3-data]’s one should be labelled [Supplementary-material SD4-data].

Reviewer #3:

This paper is a breakthrough in miRNA regulation of plant genes. Our understanding of miRNA regulation in plants has always been more advanced than animal miRNAs, mostly because the target mRNAs for plant miRNAs are so much easier to identify and study in depth. Rather than getting confused by the poorly predicted target mRNAs of animal miRNAs, the plant field has focused on the known targets and discovered amazing biology such as the phased siRNAs triggered by plant miRNAs on their target mRNAs. This paper takes another major step by purifying mRNAs and associated siRNAs from membrane fractions and finding a dramatic difference between ER polysomes and run of the mill cytoplasmic ribosome. The difference depends on how well the authors purify the various subcellular fractions and they use various compartment specific reagents to assess their purification schemes. It is convincing fractionation and the results are dramatic. The authors demonstrate step by step that AGO1 mediates the ER association of miRNAs with mRNAs, that this association is not tied to secretion of the encoded proteins, but that AGO1 is mediating the association with polysomes. And that these miRNAS cleave target mRNAs and thus trigger the production of phased siRNAs from the miRNA phase mark. The Discussion is stunningly short and to the point.

---

## [Author Response]

*Essential revisions:*

*1) Use dcl1 mutants to test whether miRNAs are required for the membrane association of AGO1. Alternatively, examine whether small RNA binding-deficient AGO1 is associated with microsomes. Small RNA binding is required for AGO1 stability. It would be very interesting if AGO1 is still associated with microsomes in the absence of miRNAs. The authors should add some discussions about this.*

We actually did examine, with many replications, mutants in a number of genes known to affect miRNA biogenesis. We included the results with *dcl1-20* in the revised manuscript. The results are shown in Figure 4 and Figure 4—figure supplement 1.

2) Examine whether the mutant form of AGO1 in ago1-27 binds miRNAs by IP-sRNA northern blot and whether it is associated with microsomes or MBPs by western blot.

We had done these experiments as well, but did not include the results as they were either negative results in that did not reveal how the membrane association of miRNAs is reduced in *ago1-27* or they were redundant with what was shown in the manuscript. To summarize the results, the mutant AGO1-27 protein still associates with miRNAs (determined by IP followed by northern blotting), which is consistent with the result showing that AGO1 IP from ago1-27 had good slicer activity, which was presented in the original manuscript as Figure 6. The binding of miRNAs by the AGO1-27 protein is now shown in the new Figure 6. The mutant AGO1-27 protein is also present in the microsome (the new Figure 4) and MBP (the new Figure 4) fractions. Therefore, as of now, we do not know how miRNA’s membrane association is reduced in this mutant.

*We have included the full reviews, which mention separate points for revisions. The experiments listed in points #3, 4 and 5 of Reviewer #1 are also nice, but after discussion, we agree that these experiments are not essential for this paper. Reviewer #2 had a few minor comments, we ask you to take them into account as well.*

*Reviewer #1:*

[…]

3) The authors detected cleavage activity/products in M and MBP fractions. This only indicates that miRNA cleavage can occur on MBPs but does not exclude the possibility that cleavage can also occur in other places. It would be nice if the authors can try to detect cleavage activity/products in MBP-depleted fractions.

This is correct. We added a sentence to talk about this.

*4) It is very interesting that ribosome occupancy may play a role in specifying a sequence for phasiRNA biogenesis. I would suggest the authors to examine the ribosome profiles of the transcripts targeted by 22-nt miRNAs that are not capable of triggering phasiRNA biogenesis.*

As shown in the manuscript, many miRNAs have 22-nt isoforms, and they do not trigger phasiRNA biogenesis. We examined the MBP ribo-seq patterns of some miRNA target transcripts, and the patterns are similar to regular protein-coding genes in that the entire ORF had ribosome footprints. We did not observe evidence for ribosome stalling before the miRNA target sites.

*5) It would be also interesting to examine whether the components in the phasiRNA biogenesis pathway including SGS3 and RDR6 are associated with M/MBP fractions.*

We agree. We plan to obtain antibodies against these proteins or transgenic lines with tags for this purpose. We do not have these resources as yet.

*Reviewer #2:*

*Post-transcriptional gene silencing can occur via transcript cleavage or inhibition of translation. Currently there is little known about the mechanism of translational inhibition in plants. This manuscript builds on previous work from the Chen lab that indicated the ER is required for microRNA-mediated translational repression. The manuscript showed differential distribution of sRNA classes in the cytosol compared with those associated with microsomes, and that microRNAs were enriched in membrane-bound polysome preparations. This membrane association in part required AGO1. Interestingly, the ago1-27 mutant, which retains AGO1 cleavage activity, showed loss of phasing of miR168-mediated target cleavage. Furthermore, and unexpectedly, TAS transcripts were found to be associated with ribosomes. Ribosome-bound regions were not uniformly distributed across these transcripts and appeared to be confined to regions upstream of the miRNA binding site. These results hint at possible new roles for membrane-bound ribosomes as a control point in sRNA production. Overall the data are novel, interesting, clear and thorough. I have only a few minor comments on the manuscript.*

*Figure 1—figure supplement 1 F: there is no reference to PEPC.*

We added a sentence to explain it in the legend.

*Figure 4. What is the sample in lane 2?*

Thank you for this question. We neglected to mention that it is “input”.

*Figure 6: Is it possible that residual cytosolic AGO1 in the microsome fraction is responsible for the observed target mRNA cleavage?*

Yes, it is possible. The fractionation is not 100% clean. The fact that the 3’ cleavage fragments can be detected in the MBP fraction (after going through two sucrose gradient fractionations) adds confidence to the claim that cleavage occurs on MBP.

*There are two [Supplementary-material SD3-data]’s one should be labelled [Supplementary-material SD4-data].*

This has been corrected.